# Assessing sensitivities of climate model weighting to multiple methods, variables, and domains in the south-central United States

Adrienne M. Wootten[1], Elias C. Massoud[2], Duane E. Waliser[3], Huikyo Lee[3]

[1]South Central Climate Adaptation Science Center, University of Oklahoma, Norman, OK, 73019, USA
[2] Computational Sciences and Engineering Division, Oak Ridge National Laboratory, Oak Ridge, TN, 37830, USA
[3]Jet Propulsion Laboratory, California Institute of Technology, Pasadena, CA, 91109, USA

*Correspondence to*: Adrienne M. Wootten (amwootte@ou.edu)

**Abstract.** Given the increasing use of climate projections and multi-model ensemble weighting for a diverse array of
applications, this project assesses the sensitivities of climate model weighting strategies, and their resulting ensemble means, to multiple components, such as the weighting schemes, climate variables, or spatial domains of interest. The purpose of this study is to assess the sensitivities associated with multi-model weighting strategies. The analysis makes use of global climate models from the Coupled Model Intercomparison Project Phase 5 (CMIP5), and their statistically downscaled counterparts created with the Localized Canonical Analogs (LOCA) method. This work focuses on historical and projected future mean
precipitation and daily high temperatures of the south-central United States. Results suggest that the model weights and the corresponding weighted model means can be sensitive to the weighting strategy that is applied. For instance, when estimating model weights based on Louisiana precipitation, the weighted projections show a wetter and cooler south-central domain in the future compared to other weighting strategies. Alternatively, for example, when estimating model weights based on New Mexico temperature, the weighted projections show a drier and warmer south-central domain in the future.
However, when considering the entire south-central domain in estimating the model weights, the weighted future projections show a compromise in the precipitation and temperature estimates. As for uncertainty, our matrix of results provided a more certain picture of future climate compared to the spread in the original model ensemble. If future impact assessments utilize weighting strategies, then our findings suggest that how the specific weighting strategy used with climate projections may depend on the needs of an impact assessment or adaptation plan.

## 1 Introduction

The simulation output from climate models has been traditionally used for research into characterizing and understanding the climate system across multiple spatial scales. In recent years, ensembles of climate projections are increasingly used for impact and vulnerability assessments (e.g., Allstadt et al. 2015; Basso et al. 2015; Pourmoktharian et al. 2016; Gergel et al. 2017; Massoud et al., 2018, 2019, 2020ab; Wootten et al., 2020ab). These include large-scale assessments, such as the
National Climate Assessment (NCA, Wuebbles et al. 2017), and local and regional assessments for individual areas of the United States. Large and local scale assessments can make use of the entire ensemble of climate projections (composed of

global climate models [GCMs]) or make use of the unweighted ensemble mean. For these assessments, using the ensemble mean provides a useful and convenient way to assess projected changes in a region. Given the coarse resolution of the GCMs (typically > 100km$^2$), many of these assessments make use of downscaled climate projections to translate larger-scale

changes to local scales.

Alongside the use of climate modeling and downscaling for climate research and increased use for impact and vulnerability assessments, there has also been a transition in the last 20 years toward using weighted multi-model means. Projections based on model weights derived from historical skill have been shown to have greater accuracy than an arithmetic multi-

model mean in many cases, provided that there is enough information to determine a weight for each model (Knutti et al. 2010; Weigel et al. 2008; Peña and Van den Dool, 2008; Min and Hense, 2006). More recently, weighting based solely on skill has given way to weighting based upon both skill and independence in recognition of differences in skill between regions and variables and the lack of independence between GCMs resulting from common bases in model structure (Massoud et al. 2019, 2020a; Sanderson et al. 2015b, 2017; Knutti, 2010; Knutti et al. 2017). In acknowledgment of studies

indicating that the global climate models are not fully independent, the Fourth National Climate Assessment (NCA4) was the first major climate assessment in the United States to use skill and independence-based model weighting on the ensemble of climate models (Sanderson and Wehner, 2017).

Several studies have examined the effect of model weighting on the outcome of climate change projections from multiple

ensembles. For example, in Massoud et al. (2019), the authors utilized information from various model averaging approaches to evaluate an ensemble from the Coupled Model Intercomparison Project Phase 5 (CMIP5; Taylor et al., 2012), finding that Bayesian Model Averaging (BMA) reduced the error by one third and constrained the uncertainty to 20-25% of the raw ensemble for projections of atmospheric river frequency. Massoud et al. (2020a) found that BMA constrained the uncertainty in precipitation projections over the contiguous United States (CONUS) to be a third of that in the original ensemble. In

Wootten et al. (2020a), the authors found that ensemble weighting can change dramatically when weighting schemes are applied to statistically downscaled ensembles compared to a raw GCM ensemble.

Other studies have applied model weighting to a certain variable or to multiple variables and went on to investigate climate change impacts for other variables (e.g., energy and hydrologic cycles). For example, Knutti et al. (2017) extended the

weighting scheme of Sanderson et al (2015a; 2017) to projections of Arctic September temperatures and sea ice, finding that the uncertainty could be constrained by the scheme while noting the proposed weighting scheme is one of several that could be used for multiple applications.  The National Climate Assessment had previously considered weighting based only on commonly used climate variables (e.g., precipitation and temperature, Wuebbles et al., 2017), but discussions to use additional variables, such as equilibrium climate sensitivity, are currently ongoing. Other studies have calculated weights

based on metrics in one domain (e.g. globally) and then applied them to projections for another domain (e.g. North America

or Europe) (Massoud et al., 2019). However, these studies are rare, as are studies providing comparisons of various weighting schemes. Examples of these studies include Shin et al. (2020), Brunner et al. (2020a), and Kolosu et al. (2021). Shin et al. (2020) suggested that researchers may provide results from several weighted ensembles to capture the uncertainties of future changes but did not explore weighting strategies beyond different weighting schemes. Brunner et al (2020a) found that the region can influence the agreement between approaches to constrain uncertainty in the CMIP5 multimodel ensemble. Finally, Kolusu et al. (2021) focusing on a water-related decision context in Africa, finds that projected risk profiles were less sensitive to the weighting schemes used. Such studies as in these examples tend to focus on the sensitivity associated with one to a few components of a multi-model weighting strategy. No prior study (to the author's knowledge) offers a comprehensive cross-comparison of the sensitivity resulting from the choices of the domain, variable, weighting scheme, and ensemble that comprise multi-model weighting strategies. In addition, the primary focus of these studies are continental regions although climate projections are now being used by regional and local organizations for climate impacts assessments and climate adaptation with additional modeling efforts.

Taking these points into consideration, we assess the choices involved with using model weighting strategies by developing and investigating a multi-dimensional sensitivity matrix for applying model averaging for the south-central region of the US. To this end, we look at mean precipitation and high temperatures as our climate variables of interest. Furthermore, we use two sub-domains, the states of Louisiana and New Mexico, alongside the south-central U.S. study region. Overall, we created and apply various sets of model weights based on several choices that are typically involved in creating a model weighting strategy: a) the choice of the ensemble (CMIP5 or downscaled), b) the choice of model weighting scheme, c) the choice of climate variable of interest (precipitation vs temperature), and d) the choice of the domain used to derive weighting (entire south-central region vs smaller sub-domain). Therefore, one example of a strategy that we apply to estimate a set of weights uses the BMA weighting method on the CMIP5 ensemble projections of the precipitation variable for the Louisiana domain. To our knowledge, there has not been a model weighting study that included as many dimensions in the experimental matrix as this study, again these are model ensemble, domain, variable, and importantly, the weighting scheme itself. Prior studies have examined some of these dimensions individually, but the comprehensive experimental matrix used here allows the comparison of modeling weighting results based on all dimensions. This is important because there could be high sensitivities in the estimated model weights based on how the weighting strategy is formulated.

Weighted multi-model means have primarily been focused on GCMs and continental scales (Brunner et al. 2019; Pickler and Mölg, 2021; Sperna Weiland et al. 2021). However, the use of climate projections has extended to regional, state, local, and tribal uses for climate impact assessments and adaptation planning. In these regional to local efforts, the raw projection data has been used but also provided to impact models (such as hydrology or crop models). Currently, impact assessments outside the traditional venues of climate modeling tend not to use weighted multi-model means but tend to use unweighted means created using downscaled GCM ensembles. Whether to use model weighting or not is currently a hot topic in the climate

modeling community, and the current study aims to comprehensively assess the sensitivity associated with multi-model ensemble weighting schemes and strategies to add further context to this debate. For reference for the reader, we define weighting schemes to refer to the numerical approach to weighting alone, such as Bayesian Model Averaging (BMA) or the approach defined by Sanderson et al. (2015, 2017). We define a weighting strategy as the weighting scheme and other choices made when using the weighting scheme to derive model weights. For example, a weighting strategy would be using the BMA weighting scheme to derive weights using the continental United States and daily high temperature alone and another weighting strategy would be using the BMA weighting scheme to derive weights using the Southern Great Plains of the United States and daily precipitation alone. Both such examples use the BMA weighting scheme, but with different choices made to derive weights, making the two examples different weighting strategies.

Our analysis results in a wide array of possible future outcomes, which comes with high uncertainties on what to expect in the future in this domain. The main question we are after is whether or not some variables or domains have projected climate change signals that have high certainty, and alternatively, we would like to find out whether or not there are climate variables in any of the regions that have highly uncertain climate change projections, and if the use of model weighting can provide a better sense of this uncertainty. We aim to address these uncertainties by applying the multi-dimensional experimental matrix of model weighting strategies and hope to inform the scientific community of these sensitivities for the benefit of future stakeholders, including climate modelers and boundary organizations providing climate services. Our purpose in this study is not to address the skill of the multi-model weighting strategies in future projections, but rather to assess under what circumstances the projections are sensitive to multi-model weighting strategies and why.

## 2 Methods and Data

### 2.1 Study Domain and Variables

The south-central United States (from about 26°N 108.5°W to 40°N 91°W) has a varied topography with a sharp gradient in mean annual precipitation from the east (humid) to the west (arid), and a generally warm climate. The Mississippi River Valley and the Ozark Mountains in the eastern portion of the region (elevations of 200–800 m), the Rocky Mountains in the west (1500–4400 m), and the Gulf of Mexico in the southeast (near sea level). Average annual precipitation in the southeast portion of the domain can be eight times higher than drier western locations and average daily high temperatures can reach 40°C (Figure 1).

### 2.2 Climate Projection Datasets

We use one member each from 26 GCMs in the CMIP5 archive to form the GCM multi-model ensemble. To form the downscaled ensemble, the same 26 GCMs are used from the downscaled projections created with the Localized Constructed Analogs (LOCA) method (Pierce et al. 2014). The LOCA-downscaled projections have been used in other studies, including

the NCA4 (USGCRP, 2017) and Wootten et al. (2020a). CMIP5 GCMs are used in this study because LOCA downscaling with CMIP6 was not available at the time of this writing. That said, the weighting schemes used here are applicable also to other ensembles such as CMIP6 and CMIP3. Therefore, the findings of this study are generalizable to other ensembles. Table S1 lists the GCMs used for both the GCM ensemble (hereafter CMIP5 ensemble) and downscaled ensemble (hereafter

LOCA ensemble). See Wootten et al. (2020a) for more details on the climate projection datasets.

To facilitate analysis, the data for each ensemble member and the gridded observations are interpolated from their native resolution to a common 10 km grid using a bi-linear interpolation similar to that described in Wootten et al. (2020b). We examine projected daily precipitation (pr) and daily high temperature (tmax) changes from 1981–2005 to 2070–2099 under

the RCP 8.5 scenario, which ramps the anthropogenic radiative forcing to 8.5 W/m$^2$ by 2100. We chose RCP 8.5 to maximize the change signals and allow us to analyze greater differences between weight schemes and downscaling techniques. The historical period (1981–2005) is used for both the historical simulations and observations to facilitate comparisons with other studies (Wootten et al. 2020b) and because the historical period of the CMIP5 archive ends in 2005 (Taylor et al. 2012).

**2.3 Observation Data**

Many publicly available downscaled projections (including LOCA) are created using gridded observation-based data for training. Gridded observations are based largely on station data that are adjusted and interpolated to a grid in a manner that attempts to account for biases, temporal/spatial incoherence, and missing station data (Behnke et al. 2016; Wootten et al. 2020b; Karl et al. 1986; Abatzoglou, 2013). In this study, we use Livneh version 1.2 (hereafter Livneh [Livneh et al. 2013]),

interpolated to the same 10 km grid using bilinear interpolation, as the gridded observation data used for comparison to the ensembles. Livneh is used in part to facilitate any comparisons between this study and the results of Wootten et al. (2020a). The LOCA ensemble used the Livneh data as the training data, so it is expected that LOCA will be more accurate than the CMIP ensemble when compared to the Livneh dataset. While we recognize that different gridded observations and downscaling techniques influence projections of precipitation variables (e.g., number of days with rain, heavy rain events),

the effect is minimal on the mean annual precipitation (Wootten et al. 2020b). Therefore, we find it is appropriate to make use of only one statistical downscaling method and one gridded observation dataset.

**2.4 Weighting Schemes**

In this analysis, we make use of model weighting schemes detailed in Wootten et al. (2020a) and similar to the weighting schemes applied in Massoud et al. (2020a). The resulting weighting schemes are applied multiple times to complete an

experimental matrix of weighting strategies allowing for in-depth comparisons of the sensitivity of the ensemble mean to various approaches to deriving and applying the multi-model weights. These weighting methods include the unweighted model mean, the historical skill weighting (hereafter Skill), the historical skill and historical independence weighting (SI-h),

the historical skill and future independence weighting (SI-c), and the Bayesian Model Averaging (BMA) method. All of the methods are calculated in the same manner as in Wootten et al (2020a). In essence, the unweighted strategy takes the simple

mean of the entire ensemble. The Skill scheme utilizes each model's skill in representing the historical observations via the root mean square error (RMSE) of the model against the historical observations. The SI-h scheme is the same weighting scheme as shown in Sanderson et al. (2017), creating an independence and skill weight using the historical simulations of each model in an ensemble. To briefly summarize the SI-h (Sanderson et al. 2017) approach, an intermodel distance matrix is calculated using the area-weighted RMSE of each model with the other models and with observations. This distance

matrix is used to calculate independence and skill weights, where the distances between one model and every other model are used to calculate the independence weight and the distance between one model and the observations are used to calculate the skill weight. The overall weight given to each model is the product of the skill and independence weights normalized such that all the overall weights for each model sums to one. The SI-c scheme is unique to Wootten et al. (2020a) and modifies the Sanderson et al. (2017) approach to use historical skill to derive the skill component of the weighting and the

climate change signal (i.e., the future projections) to derive the independence component of the weighting. To achieve this, the SI-c uses two distance matrices, the first distance matrix (used to calculate the skill weight) is the same as the SI-h, while the second distance matrix (used to calculate the independence weight) is the area-weighted RMSE of the change signals between the models. The overall weights are then calculated in the same way as the overall weights from SI-c. The BMA scheme employs a probabilistic search algorithm to find an optimal set of model weights that produce a model average that

has high skill and low uncertainty when compared to the observation and its uncertainty. BMA is an approach that produces a multi-model average created from optimized model weights, which correspond to a distribution of weights for each model, such that the BMA-weighted model ensemble average for the historical simulation closely matches the observational reference constraint. In essence, the close fit to observations is a consequence of applying higher weights on more skillful models. Furthermore, since the BMA method estimates a distribution of model weights, various model combinations become

possible, which explicitly takes care of the model dependence issue. The equations for all the weighting schemes used in this study are provided in the supplemental material, and readers are referred to Wootten et al. (2020a) and Massoud et al. (2019, 2020a) for more details on each method.

**2.5 Experimental Matrix**

Each weighting scheme (Skill, SI-h, SI-c, and BMA) is applied to both ensembles (CMIP5 and LOCA) and three domains

(south-central U.S., Louisiana, New Mexico) to fill out an experimental matrix of weights, representing a collection of weighting strategies. As a result, for each weighting scheme (skill, SI-h, SI-c, and BMA) and ensemble (CMIP5 and LOCA), there are six sets of weights produced (i.e., 3 regions and 2 variables). One example of a weighting strategy would be the BMA weighting scheme used on the CMIP5 ensemble trained on tmax for the entire domain. Another weighting strategy example would be a skill-based weighting scheme used on the LOCA ensemble trained on precipitation in Louisiana. There

are a total of 48 such model weighting strategies (ensemble choice x variable choice x weighting scheme choice x domain

choice = 2 x 2 x 3 x 4 = 48) and corresponding multi-model weights. In addition to the set of 48 weighting strategies, an unweighted ensemble mean is also used. The unweighted strategy effectively has equal weights for all models regardless of variable, domain, or ensemble. As such, including an unweighted ensemble mean represents only one additional modeling strategy, which brings the total to 49 model averaging strategies in our experimental matrix.


The various model weights from each strategy are calculated, and the derived sets of weights are then applied to create ensemble means for the three domains and two variables. In other words, a certain set of weights can be used to determine projected changes in either tmax or pr and can be used for any of the domains, the full domain, Louisiana, or New Mexico. There are a total of 288 such maps that can be created to investigate future climate change. These are 48 model averaging

choices described above, applied to 2 different variables in 3 different domains, or 48 x 2 x 3 = 288 combinations of maps. This collection of 288 is in addition to the results from unweighted means of temperature and precipitation. Including these unweighted means, there are 290 combinations of maps from this project. This explains the highly dimensional experimental matrix applied in this study, which provides the total uncertainty that is estimated with our future change projections. See Figure 2 for a schematic describing the various choices made to create each model weighting strategy and the choices made

to how each of these model weights can be applied. However, we also note that there will be several duplicates in the experiment. For example, when using the same weighting strategy, the resulting ensemble mean in a subdomain will be the same as the resulting ensemble mean in the same portion of the full domain.

## 3 Results

This section will first consider the sensitivity of the model weighting schemes to the ensembles, variables, and domains used.

This section will then focus on the bias and change signal from the resulting combinations of ensemble means.

### 3.1 Ensemble weights – results from various model weighting strategies

The resulting sets of model weights for the CMIP5 ensemble for each weighting strategy are shown in Figure 3. The 24 sets of model weights for the LOCA ensemble for each weighting strategy are shown in Figure 4. Alongside the best-estimated weight from strategies using the BMA weighting scheme, the box-whisker plots in the image show the spread of weights

from the 100 iterations of BMA for each ensemble, variable, and domain where BMA was used to derive model weights. The red dots in these figures depict the outliers from the BMA distributions of weights.

One observation is that the weighting schemes themselves are all sensitive to the ensemble, variable, and domain for which they are derived in terms of which GCMs are given the highest weight. This is reflected further when one considers which

models from each ensemble are given the strongest weights by each model weighting scheme (Table 1). From Table 1, no

model appears in the top three for all weighting strategies. The model most consistently in the top three is the CanESM2, which is in the top three for 35.4% of the 48 weighting strategies.

Although the weighting schemes are sensitive to ensemble, variable, and domain, the weights produced by Skill, SI-h, and SI-c are similar to each other, while the BMA weighting tends to be different. This is particularly true for precipitation and follows what was shown by Wootten et al. (2020a) and Massoud et al. (2020a). The BMA approach provides a distribution of weights for each model and this distribution of weights overlaps the weights of the Skill, SI-h, and SI-c approaches. This distribution of weights covers a broader region of the model weight space, but the best BMA combination (marked as orange squares in Figures 3 and 4) is noticeably different from the other schemes. The BMA best combination is the single set of

model weights from the BMA posterior that creates a weighted model average that has the best fit to the observations. Although all the samples of model weights from the BMA posterior have an improved fit compared to the original ensemble mean and provide a range of model weights as shown in the BMA distributions in Figures 3 and 4, the BMA best combination is considered the best of all these samples.

The pattern of the weights, shown in Figures 3 and 4, changes significantly between weighting strategies, particularly among the BMA weights and in the CMIP ensemble. Among the BMA and CMIP5 ensemble combinations (Figure 3), there are no common patterns to the model weights based on domain or variable. However, while the patterns between Skill, SI-h, and SI-c are similar to each other, their magnitude is consistently smaller than BMA. This indicates that when applying different weighting schemes, different models are given higher weights when applying the CMIP5 ensemble for different domains or

variables.

When using the LOCA ensemble (Figure 4), there is more consistency in which models are given higher weights, particularly for weighting strategies using high temperature (tmax). For the LOCA ensemble, the distribution of the BMA weights has a similar pattern across all three domains for the tmax derived weights, and the best-weighted models are also

somewhat consistent between domains. Similar to the CMIP5 ensemble in Figure 3, the BMA weights tend to be larger for the highest weighted models in the LOCA ensemble compared to those derived with the Skill, SI-h, and SI-c schemes. We speculate that the reason for this is because the Skill, SI-h, and SI-c schemes involve the 'skill' of each model when estimating weights, and since the LOCA downscaled ensemble is bias corrected, most models have similar skill and therefore similar weights. For weights derived with tmax, the Skill, SI-h, and SI-c have very similar patterns for both the full

and New Mexico domains. The Skill and SI-h weighting schemes, which focus entirely on the historical period, created nearly identical weights for the 26 models when weights are derived based on tmax in the full and New Mexico domains. While the weights from Skill and SI-h are not identical when derived using tmax in the Louisiana domain, the weights for the LOCA ensemble in Louisiana generally range from 0.025 to 0.050. The SI-c weights derived using tmax in the LOCA ensemble have a similar pattern between the full and New Mexico domains, but a very different pattern in the Louisiana

domain (Figure 4). In addition, the SI-c also tends to have a different pattern from the Skill and SI-h weights when tmax and LOCA are used for derivation. There is much more sensitivity to domains when using precipitation and the LOCA ensemble to derive weights, compared to that of tmax. Regardless of the weighting scheme, there is no common pattern in the weights between domains when the LOCA ensemble and precipitation are used to derive weights. Again, the BMA scheme applies much larger weights to the top models for precipitation-based LOCA weighting compared to the Skill, SI-h, and SI-c

weighting schemes.

The LOCA statistical downscaling method, like most statistical downscaling methods, incorporates a bias correction approach, which inherently improves the historical skill. In addition, the Skill, SI-h, and SI-c methods focus primarily on the first moment of the ensemble distribution when deriving weights, which limits the ability to penalize for co-dependence

between models in an ensemble. Finally, the BMA considers multiple moments of the ensemble distribution using multiple samples via Markov Chain Monte Carlo (MCMC), rewarding skillful models and penalizing co-dependency. Of the weighting combinations used here, the BMA tends to be the most sensitive to the ensemble, variable, and domain used to determine weights. Given that the BMA focuses on multiple moments of the distribution and is most sensitive to the different choices considered here (ensemble, variable, and domain) it is plausible that the BMA approach responds to and

captures the changes in skill and co-dependence among the ensemble members resulting from these various choices.

### 3.2 Size of the experimental matrix of model weights and how to apply them

One can apply the 48 weighting strategies described above in a similar manner to the way the weighting strategies themselves are created. For example, one could apply the weights derived from the CMIP5 ensemble precipitation for the full domain using BMA to create a weighted ensemble mean of CMIP5 precipitation for Louisiana. As shown in Figure 2,

each weighting strategy is applied to the variables (high temperature and precipitation) and domains (full, Louisiana, and New Mexico) to produce a set of ensemble means. Altogether, the maximum number of weighted ensemble means produced with these 48 weighting strategies is 48x2x3=288. However, this maximum number of ensemble means resulting from the experiment contains several duplicates. For example, when using the same set of weights, the resulting ensemble mean in a subdomain will be the same as the resulting ensemble mean from the same portion of the full domain. As such, the actual

number of ensemble means in this experiment is smaller than 288.

### 3.3 Historical Bias and Future Projected Changes in unweighted model ensembles

The figures shown in later sections focus on the ensemble means from the 48 weighting strategies applied to the full domain. The discussion surrounding bias and projected changes represented by the ensemble means in the following subsection will be compared to the unweighted ensemble means of high temperature and precipitation from the CMIP5 and LOCA

ensembles. For this reason, we first show the historical ranges and the ranges of the future projected changes using the unweighted model ensemble (Figure 5) before reporting on the results using the weighted ensembles. The unweighted

CMIP5 ensemble as a whole tends to underestimate high temperatures in the historical period, overestimate precipitation in New Mexico, and underestimate precipitation in Louisiana (top left panel of Figure 5). The LOCA ensemble is much closer to the Livneh observations, which is expected given the bias correction applied in statistical downscaling. Yet, for the unweighted LOCA ensemble, there is a tendency to underestimate precipitation in the whole domain and the New Mexico subdomain and to overestimate temperature in all of the domains (bottom left panel of Figure 5). For the future projected changes in the unweighted CMIP and LOCA ensembles, the projected high temperature changes are consistent between ensembles (bottom right panel of Figure 5), and the projected changes in precipitation are less variable in the LOCA ensemble for the New Mexico domain and more variable for the Louisiana domain (top right panel of Figure 5). In addition, the right-hand panels of Figure 5 show that the projected changes around the mean from the raw ensemble are significantly larger than the reduced spread in Figure 6 (particularly from the BMA results) in the weighted ensembles. This suggests that the raw ensemble has less confidence for both variables, both ensembles, and all three regions compared to the weighted ensembles. Given this baseline information, the following subsections discuss and compare the unweighted and weighted ensemble means for each ensemble (CMIP5 and LOCA). The weights for each model from each multi-model weighting strategy are given in Tables S2-S7.

### 3.4 Historical Bias and Future Projected Changes using the weighted ensembles

The 48 weighting strategies are then applied across three domains and two variables to produce 288 ensemble means. The mean projected changes can be sensitive to the weighting scheme, domain, and variable used. The future projected changes from the different ensemble means are summarized in Figure 6, where the boxplots represent the range of the ensemble mean change from the 100 BMA posterior weights. When the weighting strategy uses tmax, the resulting CMIP5 mean projected change shows predominantly a decrease in precipitation for all domains (top-left group of panels in Figure 6, top row of figures). For the weighting strategies using tmax with the LOCA ensemble (top right group of panels in Figure 6, top row of figures), the mean precipitation projections are more variable concerning the domain the weighting is applied.

Using weighting strategies using precipitation and the CMIP5 ensemble, the mean projected precipitation increases/decreases when Louisiana/New Mexico is used to derive weights across all three applied domains (top-left group of panels in Figure 6, bottom row of figures). For weighting strategies using precipitation in the LOCA ensemble, the mean projected precipitation generally decreases for most weighting schemes (top right group of panels in Figure 6, bottom row of figures), except for the resulting means for Louisiana with the BMA weighting scheme. In contrast to precipitation, the ensemble mean changes for tmax are fairly consistent for both CMIP and LOCA ensembles (bottom groups of panels in Figure 6, all rows of figures), with all model weighting strategies indicating a consistent increase in temperature for all domains.

As for the uncertainty in the results, we find in our matrix of results a reduction in the overall uncertainty compared to the spread in the original ensemble. This can be seen when comparing the results of the unweighted (Figure 5) and weighted ensembles (Figure 6). Although the maps of future change and the results from Figure 6 show that the weighted ensemble means have different results based on the weighting strategy used, the overall uncertainty is still reduced when applying model weighting even when considering the many strategies implemented in this study. This is particularly evident when examining the results for those strategies using the BMA weighting scheme (Figure 6).

Aside from the comparisons of the weighted mean change to the raw ensemble change and unweighted mean change, one can consider the magnitude of these means compared to the internal variability of the climate models and intermodel spread of the projected change. The intermodel spread calculated here is represented by the unweighted standard deviation of the projected change of ensemble members. The internal variability of the historical and future period is represented by the ensemble average of the standard deviation of each variable from each ensemble member over time (per Hawkins and Sutton, 2009; 2011, Maher et al. 2020) for each of the three domains (full, Louisiana, and New Mexico). However, we note that the forcing response is not removed given the temporal period is not continuous which is a caveat for this analysis. In the case of tmax, the projected changes from each ensemble mean is greater than the internal variability of the models and the intermodel spread regardless of the weighting scheme, ensemble, domain used to derive the weights, or the variable used to derive the weights (Figure 7). In contrast, the differences between weighting strategies do result in some differences in weighted means for the projected change in precipitation that are comparable to the internal variability and intermodel spread. For example, for the CMIP5 ensemble means weighted for Louisiana precipitation and applied to Louisiana precipitation, the difference between the BMA ensemble mean and the unweighted mean is comparable to the intermodel spread and internal variability. In addition, the difference between the BMA ensemble mean created based on Louisiana precipitation and all the weighted ensemble means created based on full domain precipitation is also comparable to the intermodel spread and internal variability. Overall, results in Figure 7 suggest that, in general, the projected changes in temperature are larger than the ensemble spread and the internal variability of temperature, whereas for precipitation, the projected changes are not as great as the original ensemble spread or the internal variability of precipitation.

The following section and corresponding figures compare the results from the various weighting strategies applied in this study. Figure 8 looks at historical biases and Figure 9 shows the projected future change signals in precipitation for the for strategies using the CMIP5 ensemble. Figures 10 and 11 look at historical bias and projected future change signals in high temperature for CMIP5. Figure 12 looks at the projected future change signal in precipitation for weighting strategies using the LOCA ensemble, and Figure 13 looks at the projected future change signal in high temperature for weighting strategies using the LOCA ensemble. For an in-depth analysis of how the model weighting strategies impact the resulting historical bias and climate change signals shown in Figures 8-13, readers are referred to the supplementary section, with a discussion on the main findings reported in the next section. For additional results that complete the analysis, readers are referred to the

supplementary section (Figures S1-S6), which includes bias maps from the LOCA ensemble (S1-S2) as well as error distributions from the historical simulations of both ensembles (S3-S6). Figures S3-S6 indicate that all the weighting

strategies used in this study resulted in higher skill for both high temperature and precipitation in all three domains. To summarize the results for skill, the RMSE of each weighting strategy is shown for all three domains for precipitation and high temperature in Table 2 and Table 3 and the RMSE for the unweighted cases are in Table 4. Of the weighting strategies using the CMIP5 ensemble 92%, 92%, and 75% have lower RMSE for precipitation than their unweighted counterparts for the full, New Mexico, and Louisiana domains. Similarly for the high temperature, 96%, 100%, and 79% of weighting

strategies have lower RMSE than their unweighted counterparts for the full, New Mexico, and Louisiana domains. Therefore, most weighting strategies have higher skill than the unweighted CMIP5 ensemble. However, there is a similar pattern for weighting strategies using the LOCA ensemble. For precipitation, 79%, 58%, and 67% of weighting strategies using the LOCA ensemble have a lower RMSE than their unweighted counterparts for the full, New Mexico, and Louisiana domains. Similarly for high temperature, 88%, 88%, and 83% of weighting strategies using the LOCA ensemble have a

lower RMSE than their unweighted counterparts. It is important to note that this analysis of RMSE and bias is for the historical period only. Prior studies have noted that reducing historical biases does not mean better performance during the future period (Dixon et al. 2016; Sanderson et al. 2017). Therefore, historical skill alone does not justify the use of any weighting strategy. In what follows, we do not recommend using any specific weighting strategy based on the historical skill. Rather, we focus on the sensitivity of the projected changes to the various weighting strategies.

**4 Discussion**

Among climate scientists and the climate modeling community, there is a debate regarding the weighting of multi-model ensembles and, if one does apply weighting, how to do so. This debate includes scientists involved in the development of climate projections for the United States' Fifth National Climate Assessment (US 5[th] NCA report), as well as other national and international assessments. The authors of this study are involved in the development of climate projections for the US 5[th]

NCA report via group discussions on climate modeling, downscaling, and model weighting, and these discussions include the same questions of interest in this study. The debate over climate model weighing, particularly as connected with the NCA, is a main reason that this study investigates an extensive and comprehensive research matrix. Previous studies, such as those of Sanderson et al (2015 and 2017) and Knutti (2017) have focused on the evaluation and application of singular weighting strategies, while other studies have begun to consider the added components of bias correction (Shin et al. 2020),

additional approaches to weighting (Brunner et al. 2020b), and the sensitivities of multi-model ensemble weighting in small regions (Kolusu et al. 2021).

This is the first study, to the authors' knowledge, to comprehensively assess the sensitivities of the model weights and resulting ensemble means to the combinations of variables, domains, ensemble types (raw or downscaled), and weighting

schemes used for a large and complex region of the United States. The specific weighting schemes used include the Sanderson et al. (2017) approach and the Bayesian Model Averaging (BMA; Massoud et al. 2019, 2020a; Wootten et al. 2020a). The former approach is a prominent weighting scheme used in the Fourth National Climate Assessment, while the BMA is an increasingly prominent technique that will be used to create the projections in the Fifth National Climate Assessment (NCA). The remaining two weighting schemes used are a variation of the Sanderson et al. (2017) method

proposed by Wootten et al (2020a) and a common skill weighting approach. These weighting schemes are compared alongside the resulting values from an unweighted ensemble mean, which is the most commonly used from of multi-model ensemble averaging in the literature. Therefore, this study quantifies multiple weighting sensitivities to inform the larger discussion on multi-model ensemble weighting. Our study assesses the sensitivities associated with multi-model weighting strategies but does not consider the skill of the model-weighting strategies in the future projections. This latter aspect is the

subject of future work.

## 4.1 Sensitivities of the Results to the Experimental Design

The results from individual weighting schemes are sensitive to the choice of domain and variable of interest, regardless of whether the ensemble is downscaled or not. However, one can also note that the BMA weighting scheme tends to be more sensitive than the others. As noted by Wootten et al. (2020a) and Massoud et al. (2019, 2020a), the Skill, SI-h, and SI-c

weighting schemes focus on the first moment of the distribution of a variable, while the BMA approach focuses on multiple moments of the distribution of weights. The BMA weighting can therefore produce weights that are significantly different from the other weighting schemes. In addition, the BMA will also be more sensitive to the differences between domains and variables that are provided to derive model weighting. This is particularly the case with regards to the CMIP5 ensemble results for both variables but also is evident in the LOCA ensemble results for precipitation. The ensemble weights are most

sensitive to the variable and domain using the CMIP5 ensemble and the weights created with the LOCA ensemble are less sensitive. A statistical downscaling procedure reduces the bias of the ensemble members compared to the raw CMIP5 ensemble, which likely results in there being less sensitivity when the LOCA ensemble is used. This is particularly likely for high temperatures, which is traditionally much less challenging for both global models and downscaling techniques to capture.


We find that, for precipitation, the ensemble mean projected change from a multi-model ensemble is sensitive to the various choices associated with the derivation of model weighting. In contrast, for high temperature, the ensemble mean projected change is less sensitive. We also find that, while a weighting strategy offers greater skill than an unweighted ensemble mean, there are distinct difference based on the ensemble, variable, and domain of interest. The larger domain of the south-central

region contains multiple climatic regions. The western portion of the domain includes the arid and mountainous New Mexico and Southern Colorado. The eastern portion of the domain is the much wetter and less mountainous area of Louisiana, Arkansas, and southern Missouri. The complexity of the region presents a challenge to GCM representation of precipitation

and temperature. Deriving ensemble weights based on Louisiana precipitation favors models which are wetter while deriving ensemble weights based on New Mexico precipitation favors those models which are drier. This effect translates into the

projected changes for precipitation in the CMIP5 ensemble that can reverse the change signal in the domain (Figure 9). The sensitivity for precipitation is evident when precipitation is the focus for deriving model weights, but also present to a lesser degree when high temperature is the focus for deriving model weights. The high temperature changes are also sensitive to the domain when precipitation weighting is used because precipitation-based weighting favors wetter or drier models (Figure 11). In contrast, the high temperature change from the CMIP5 ensemble is much less sensitive when calculated with weights

derived from high temperatures. The sensitivity present using the CMIP5 ensemble is less apparent for the projected changes with the LOCA ensemble. LOCA ensemble means derived using the BMA weighting are more sensitive to the variable and domain used to derive weights. The LOCA downscaling, like most statistical downscaling methods, corrects the bias of the CMIP5 ensemble, pushing all models to have similar historical skill. It follows that the BMA weighting is more sensitive to the different choices considered here (ensemble, variable, and domain) and that the BMA weighting responds to and captures

changes in skill and co-dependence resulting from the different options of ensemble, variable, and domain. One caveat in this study is that the sub-domains of New Mexico and Louisiana are small compared to the resolution of the GCMs in CMIP5. This suggests that natural variability may have had some effect on the results. In future work, the authors will repeat this analysis using the larger regions of the United States used in the National Climate Assessment.

**4.2 Consideration of weighting scheme, variables of interest, and domain choice**

This study finds that mean projections of temperature are much less sensitive to the weighting scheme used, while mean projections of precipitation are more sensitive, particularly if the domain is very humid or very arid (Figures 8-13). The use of multiple weighting strategies would allow for the sensitivities associated with model weighting to be captured and considered.

The results from this study also suggest that weighting on specific variables could be used to address the large biases and co-dependencies with respect to that variable among the models and produce ensemble means that reflect the appropriate confidence with regards to that variable. However, temperature, precipitation, and multiple other variables have strong physical relationships and thus are not fully independent themselves. As such, creating separate weights for variables independently may break the physical relationships between variables in resulting ensemble means. In addition, this study

did not examine multivariate weighting strategies such as the implementation of SI-h in the Fourth National Climate Assessment (Sanderson and Wehner. 2017). A multivariate weighting strategy (a weighting scheme used with multiple variables in a given domain) will likely retain physical relationships between variables when used to calculate a multi-model ensemble mean. However, this was not explored in this study. In addition (to our knowledge), the sensitivity of multivariate weighting strategies has not been explored in prior literature. Future work will explore multivariate ensemble weighting in

greater depth.

Climate model evaluations and national and international assessments typically focus global or continental areas. However, the individual National Climate Assessment (NCA) regions are climatically very different from each other. The individual GCMs in the CMIP ensemble likely do not have the same performance across all regions and an individual downscaling technique can be evaluated in one of these regions but applied to the entire continental United States or North America. In addition, the regions of Alaska, the U.S. Pacific Islands, and the U.S. Caribbean Islands have vastly different climates to the continental United States. In this study, we have found that the weighting for precipitation in particular can be very sensitive to domain and weighting scheme used. This is also found to be the case to a lesser degree with temperature. Based on this study, the model weighting for each of the NCA regions will likely be vastly different than the weighting for the continental United States as a whole.

At the time of writing, discussion surrounding the use of weighted multi-model ensembles has been traditionally limited to climate model developers and the production of national or international climate assessments but is beginning to be used in impact assessments. Among climate model developers, Knutti et al. (2017) argue that model weighting is a necessity in part to account for situations where the model spread in the present-day climatology is massive resulting in some models having biases so large that using an unweighted mean is difficult to justify. In other situations, model interdependence becomes increasingly relevant with the increased use of common code bases across institutions causing unweighted means to be overconfident (Brunner et al., 2020b). This concern was also shared by Wootten et al. (2020a) with respect to the common modeling code base applied in the statistical downscaling process. Based on expert discussions surrounding downscaling and model weighting, the NCA is now considering weighting based on model climate sensitivity as opposed to traditional model weighting approaches (Nijsse et al., 2020; Hausfather et al., 2022). This study demonstrates that the weights and resulting ensemble means (particularly for precipitation) are sensitive to the ensemble (CMIP or LOCA), variable, and domain used. However, nothing done in this study negates the concerns of Knutti et al. (2017) and Wootten et al. (2020a). An unweighted mean will allow models with large biases and co-dependencies regardless of the domain or variable of interest larger influence in either climate models or impact assessments. Therefore, although this study demonstrates that resulting ensemble means for variables of interest are sensitive to the choice of weighting strategy, a weighting strategy should still be used, with careful consideration given to domain, variable, and weighting scheme used.

**4.3 Challenges and Future Work**

In this study, we showed that the weighting schemes and the resulting weighted ensemble means are sensitive to the domain and variable used. We have several findings from this analysis. Firstly, we find that the model weights themselves are sensitive to the weighting scheme used, with BMA being the most sensitive owing to the ability to capture multiple moments

of the distribution of the ensemble. Second, we find that precipitation can be highly sensitive to the domain used in the weighting strategy. We also find that temperature can also be sensitive to a lesser degree.


Our findings are somewhat different from those of Befort et al. (2022) which suggests the impact of model weighting strategy to be minor. However, we note that Befort et al. (2022) focused on decadal climate prediction with a smaller ensemble than the one presented in this study. At the time of this writing, some intercomparisons of weighting schemes have been attempted in smaller regions with a smaller number of weighting strategies. Balhane et al. (2022) found that resulting

model weights are sensitive to the quantity of interest used in derivation while also finding that model weights are less sensitive to spatial domain. This agrees in part with our study showing the model weighting and resulting means are sensitive to both the domain and variable of interest used with the model weighting scheme. A future application of this analysis may incorporate additional concerns or new approaches to ensemble weighting associated with emergent constraints. The relationship between interannual variability and long-term trends has led to new consideration of weighting an ensemble by

the ability of individual models to represent observed variability for the quantities of interest (Wenzel et al. 2014; Nijsse et al., 2020; Balhane et al. 2022). As described here, this reflects changing the variable of interest in a weighting strategy to reduce the uncertainty in the ensemble by constraining it to the observed variability. While the emergent constraints approach is potentially useful, it depends upon having reliable observations of each variable and thorough understanding of the physical process alongside multiple other limitations (Kuepp et al. 2019; Caldwell et al. 2018). In addition, the emergent

constraints approach focus on observed variability may not address the "hot model" problem identified by Hausfather et al. (2022), where models with overly high climate sensitivity result in overly large projected increases in global temperatures. While neither approach were the focus of this study, future research should consider the impacts of using one or both (observed variability or climate sensitivity) in weighting strategies to constrain climate model ensembles and reduce uncertainty. In this study, we focused on the sensitivity under RCP 8.5 to maximize the effects observed from different

weighting strategies. Given the smaller change signals under other RCPs it is possible that the sensitivities observed here have a lesser magnitude under other RCPs. Considering this component is another aspect that could be explored in future work.

Finally, the authors recognize that the climate modeling community and connected stakeholders are incorporating climate

model simulations as inputs to additional modeling efforts such as hydrology modeling or crop modeling for use in impacts assessments. While most impact assessments have not incorporated model weighting directly, some studies are beginning to do so (e.g., Skahill et al., 2021; Amos et al. 2020; Sperna Weiland et al. 2021; Schäfer Rodrigues Silva et al. 2022; Elshall et al. 2022). There are known non-linear relationships between climate and impacts modeling (such as hydrology or crop modeling). Would weighting strategy that used climate model inputs produce the same result as multi-model weighting

based on, for example, streamflow output using an ensemble of climate projections as inputs? Given the sensitivities associated with weighting schemes, variables, domains, and ensembles (identified in this study), we suspect that the

weighting strategy would not be the same when using the output of an impacts model (such as streamflow) and that the translation of error and co-dependencies from climate model projections to impacts models (may result in a higher degree of sensitivity with respect to stakeholder specific variables (such as streamflow). Therefore, the questions of sensitivity of

weighting strategies and ensemble means bear increasing relevance as the number of users of climate projection output continues to increase.

## 5 Conclusions

This study examines the sensitivity of the multi-model ensemble weighting process and resulting ensemble means to the choices of variable, domain, ensemble, and weighting scheme for the south-central region of the US. In general, we see that

weighting for Louisiana makes the future wetter and less hot, weighting for New Mexico makes the future drier and hotter, and accounting for the whole domain provides a compromise between the two. In addition, we see that ensemble mean projections for precipitation are more sensitive to the various aspects tested in this study, while ensemble mean projections for high temperature are less sensitive. As such, some domains/variables have uncertain outcomes, regardless of the weighting method. But for other domains/variables, the uncertainty is dramatically reduced, which can be helpful for the

assessment of climate models and climate adaptation planning. The sensitivity of precipitation and temperature projections is reduced when LOCA is used, which is likely the result of the bias correction associated with the LOCA downscaling method. In addition, the BMA weighting scheme is more sensitive than the other weighting schemes. BMA's sensitivity is the result of the BMA approach focusing on multiple moments of the distribution to account for model biases and co-dependencies.


Although there is sensitivity associated with the model weighting, efforts using a multi-model ensemble of climate projections should incorporate model weighting. Model weighting still accounts for issues of bias and co-dependence that preclude a model democracy approach to crafting multi-model ensemble means. Incorporating multiple weighting schemes allows for assessing and capturing the sensitivity associated with model weighting to the benefit of both climate modeling

efforts and climate adaptation efforts. Given the sensitivity associated with weighting for different variables and domains, one may also consider crafting weighting schemes with a focus on the domains or variables of interest to an application. In addition, since some impact assessments or adaptation planning efforts make use of climate projections as inputs to impacts models (such as hydrology or crop models) there is a need to consider similar research to this study with regards to the direct outputs of impacts models using climate projections.


There are a couple of caveats and suggested future research. First, this study makes use of domains that are fairly small where the spatially aggregated internal climate variability is larger than that of a large domain. Second, this study focused on the south-central United States. Future efforts should consider this analysis using larger regions, such as the continental

United States and the NCA sub-regions. Future efforts should also consider examining multivariate weighting to account for the physical relationships between variables. Third, this study does assume stationarity in the multi-model ensemble weights and resulting weighted means. Future research will examine the accuracy and sensitivity using a perfect model exercise (such as what is described by Dixon et al. 2016 and Sanderson et al. 2017) to test the stationarity assumption associated with ensemble weighting. This is important since studies like Sanderson et al. (2017) show that a more skillful representation of the present-day state does not necessarily translate to a more skillful projection in the future. Our study does not consider the skill of the multi-model weighting strategies in the future projections, but rather it assesses the sensitivity of future projections to the various multi-model weighting strategies. Fourth, this study did not consider an emergent constraints approach on either observed variability or climate sensitivity, which should be considered in future research. Finally, given the increasing use of climate model ensembles in impacts models, future efforts should consider a similar investigation to this study using an impacts model. Such future efforts will answer multiple questions regarding the appropriate model weighting schemes, but also provide potential guidance to boundary organizations building capacity to assist in regional and local climate adaptation planning and impact assessments.

**6 Code Availability**

R Code to calculate weights associated with the Skill, SI-h, and SI-c weighting and produce all analysis in this study are available from Dr. Wootten on request. Programming code for BMA calculations is available from Dr. Massoud on request.

**7 Data Availability**

CMIP5 GCM output are available through the Earth System Grid Federation Portal at Lawrence Livermore National Laboratory (https://esgf-node.llnl.gov/search/cmip5/). The LOCA downscaled climate projections for CMIP5 GCMs are available through numerous portals included the USGS Center for Integrated Data Analytics GeoData Portal (cida.usgs.gov/gdp). The Livneh gridded observations are available from the National Centers for Environmental Information (https://www.ncei.noaa.gov/access/metadata/landing-page/bin/iso?id=gov.noaa.nodc:0129374;view=html).

**8 Author Contribution**

Dr. Wootten and Dr. Massoud – Conceptualization, Formal Analysis, Investigation, Methodology, Writing – original draft preparation, Writing – review and editing, Visualization, Validation. Dr. Wootten – Data Curation. Dr. Waliser and Dr. Lee – Supervision, Writing – review and editing.

## 9 Competing Interest

The authors declare that they have no conflict of interest.

## 10 Acknowledgements

A part of the research was carried out at the Jet Propulsion Laboratory, California Institute of Technology, under a contract with the National Aeronautics and Space Administration (80NM0018D0004). The authors thank the reviewers for their comments and critiques to strengthen this article. This manuscript has been authored by UT-Battelle, LLC, under contract DE-AC05-00OR22725 with the US Department of Energy (DOE). The US government retains and the publisher, by accepting the article for publication, acknowledges that the US government retains a nonexclusive, paid-up, irrevocable, worldwide license to publish or reproduce the published form of this manuscript, or allow others to do so, for US government purposes. DOE will provide public access to these results of federally sponsored research in accordance with the DOE Public Access Plan.

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

**Figures**

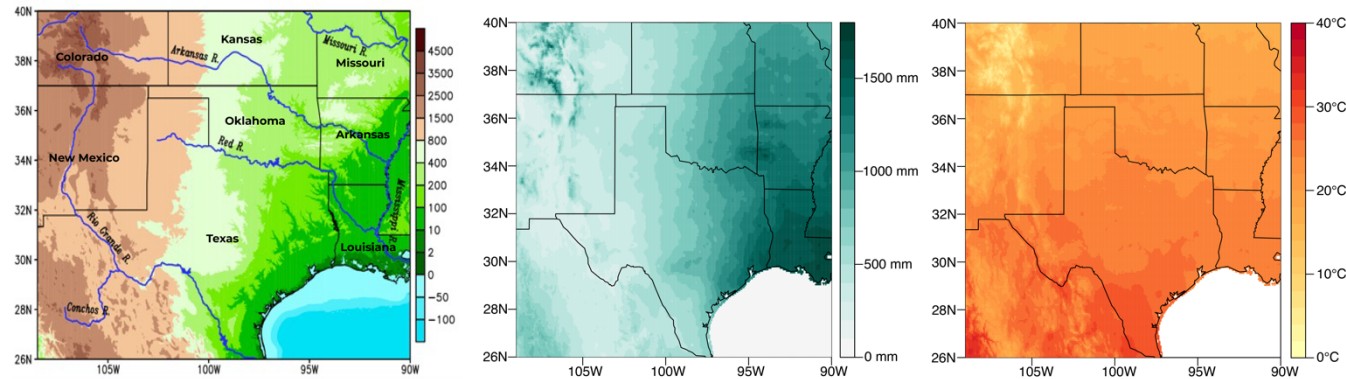

**Figure 1: Topographical map for the study domain: The elevation map of the south-central United States with major rivers overlaid on it. Brown/green shading denotes elevation (in units of m), while the rivers are outlined in blue. Topography, bathymetry, and shoreline data are obtained from the National Oceanic and Atmospheric Administration (NOAA) National Geophysical Data Center's ETOPO1 Global Relief Model (Amante and Eakins, 2009). This is a 1 arc-minute model of the Earth's surface developed from diverse global and regional digital datasets and then shifted to a common horizontal and vertical datum. River shapefiles are obtained from the Global Runoff Data Centre's Major River Basins of the World (GRDC 2020). Center — Study domain overlaid with annual average precipitation (mm) from Livneh v. 1.2 (Livneh et al. 2013). Right— Study domain overlaid with annual high temperatures (°C) from Livneh v. 1.2 (Livneh et al. 2013).**

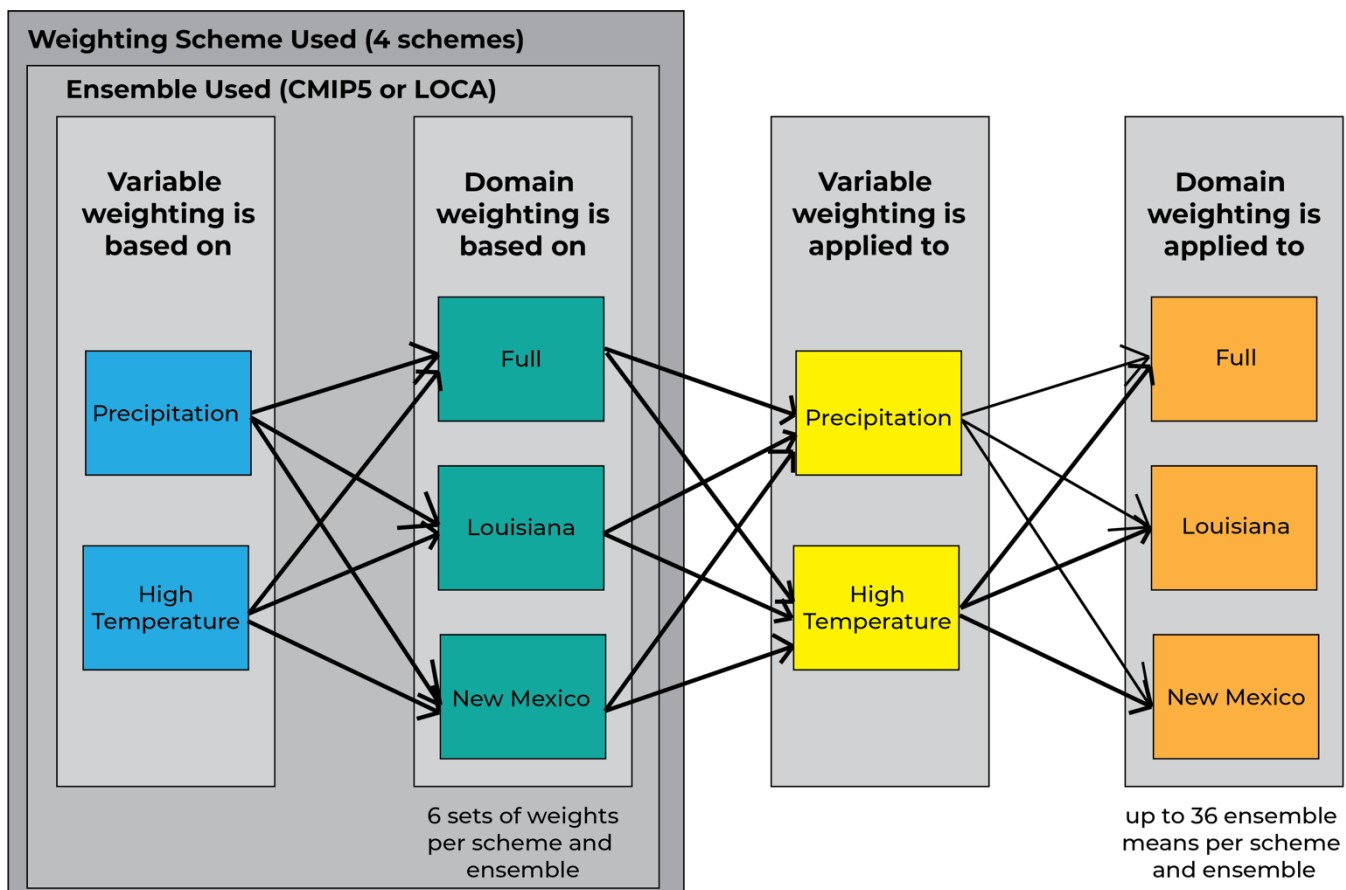

**Figure 2: Flowchart showing the process of analysis with weighting schemes. Each version of the model average is constructed based on several choices: a) the choice of the ensemble (CMIP vs LOCA), b) the choice of model weighting strategy (unweighted, Skill, SI-h, SI-c, or BMA), c) the choice of climate variable of interest (precipitation or temperature), and d) the choice of the domain used for the ensemble averaging (entire south-central region, Louisiana, or New Mexico). These various choices give up to 48, plus the unweighted version, so 49 overall choices of model weighting strategies. Then, once the model average is constructed**
**and trained, there is a choice to be made on which variable and which domain to apply this model average to. Therefore, this results in 48 x 2 x 3 = 288 possible future outcomes in our experimental matrix plus 2 unweighted outcomes, for a total of 290 combinations.**

# CMIP5 Ensemble Weights

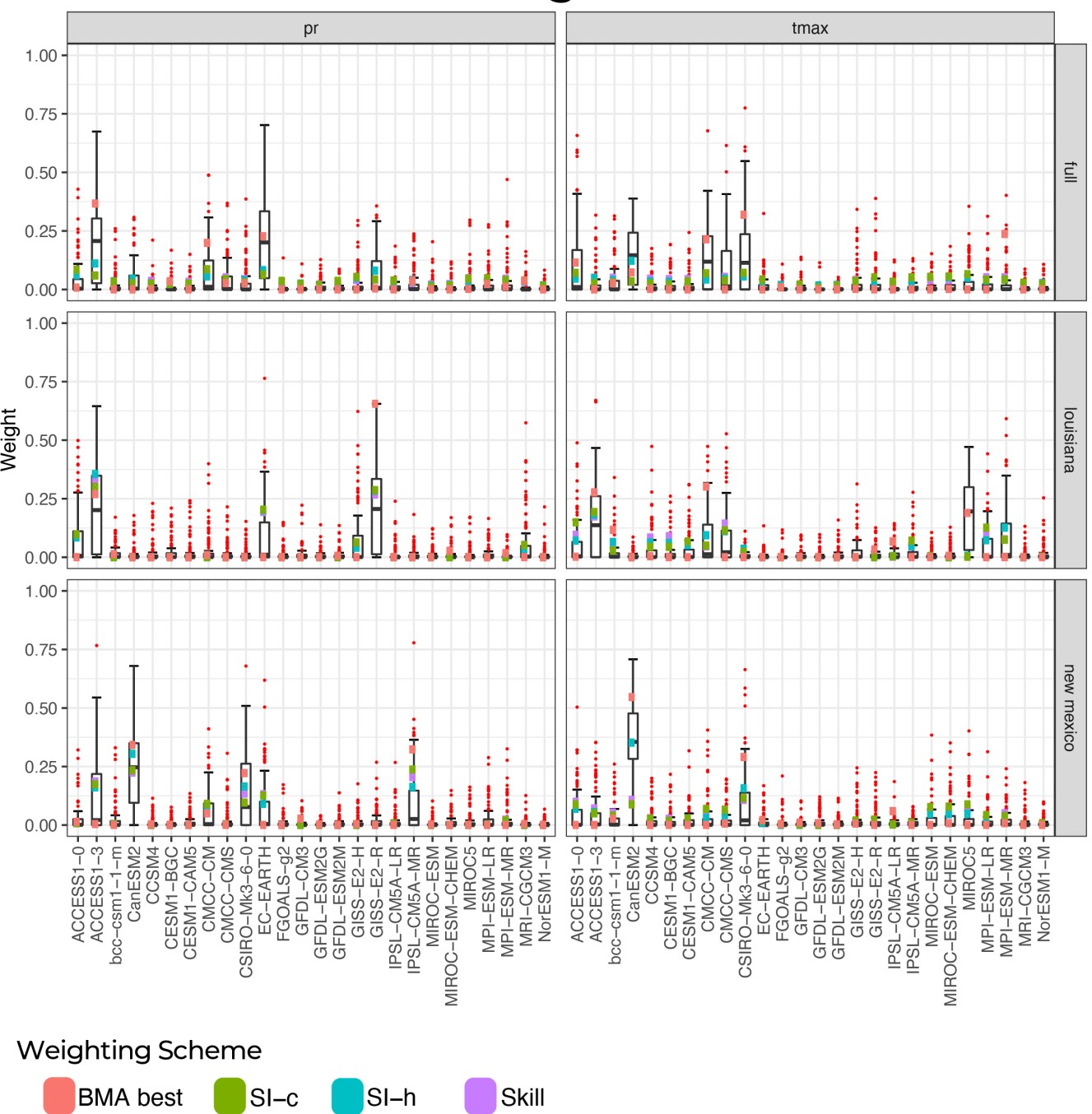

**Weighting Scheme**

■ BMA best   ■ SI–c   ■ SI–h   ■ Skill

**Figure 3: Model Weights for each of the 4 weighting schemes using the CMIP5 ensemble. The left column is weights based on precipitation (pr) alone and the right column is weights based on high temperature (tmax) alone. The top row is weights based on the full domain, the middle row is weights based on Louisiana alone, the bottom row is weights based on New Mexico alone. The boxplots are the spread of weights from the 100 iterations of the BMA weighting scheme. The red dots in these figures depict the outliers from the BMA distributions of weights.**


# LOCA Ensemble Weights

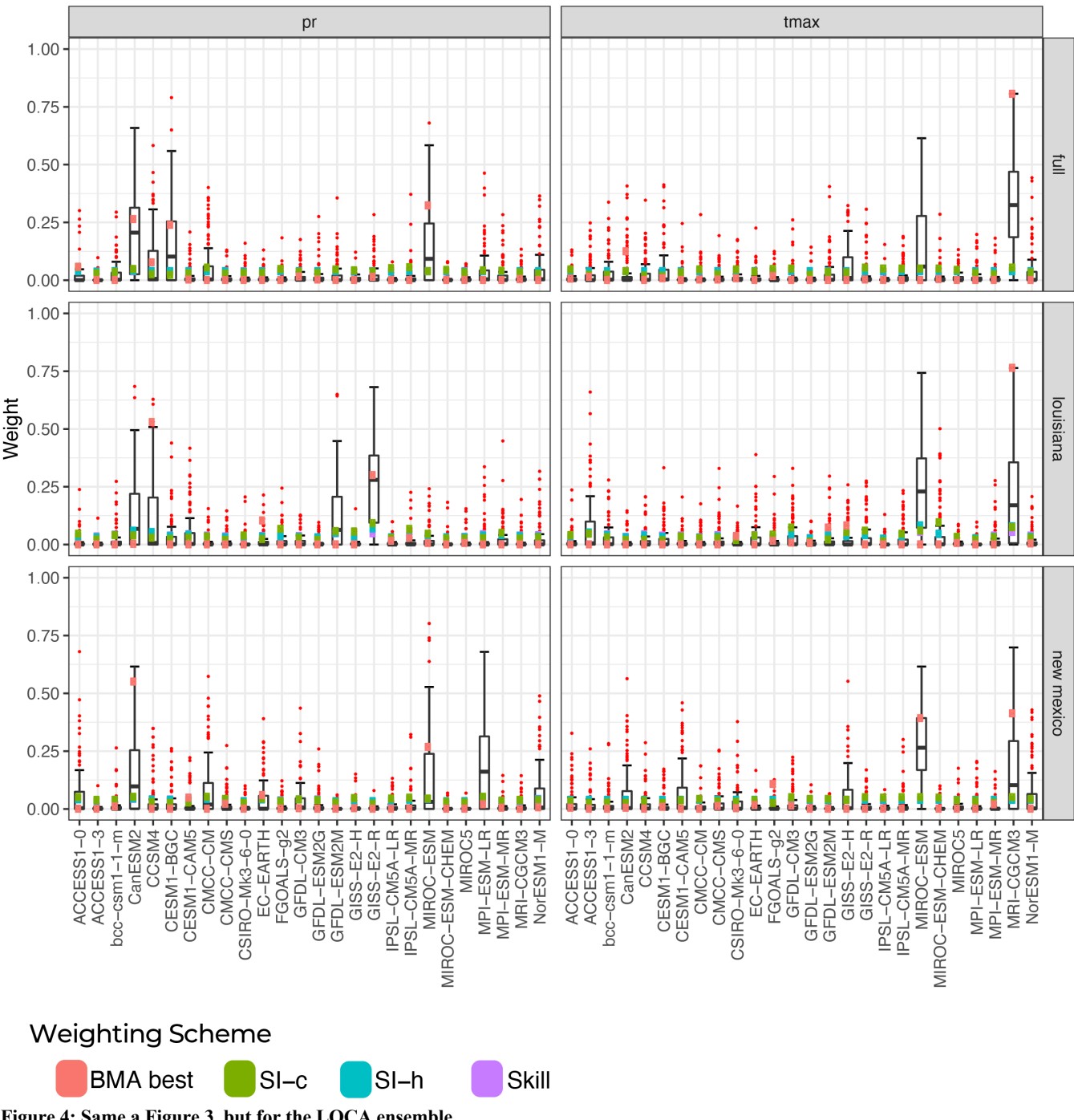

**Figure 4:** Same a Figure 3, but for the LOCA ensemble.

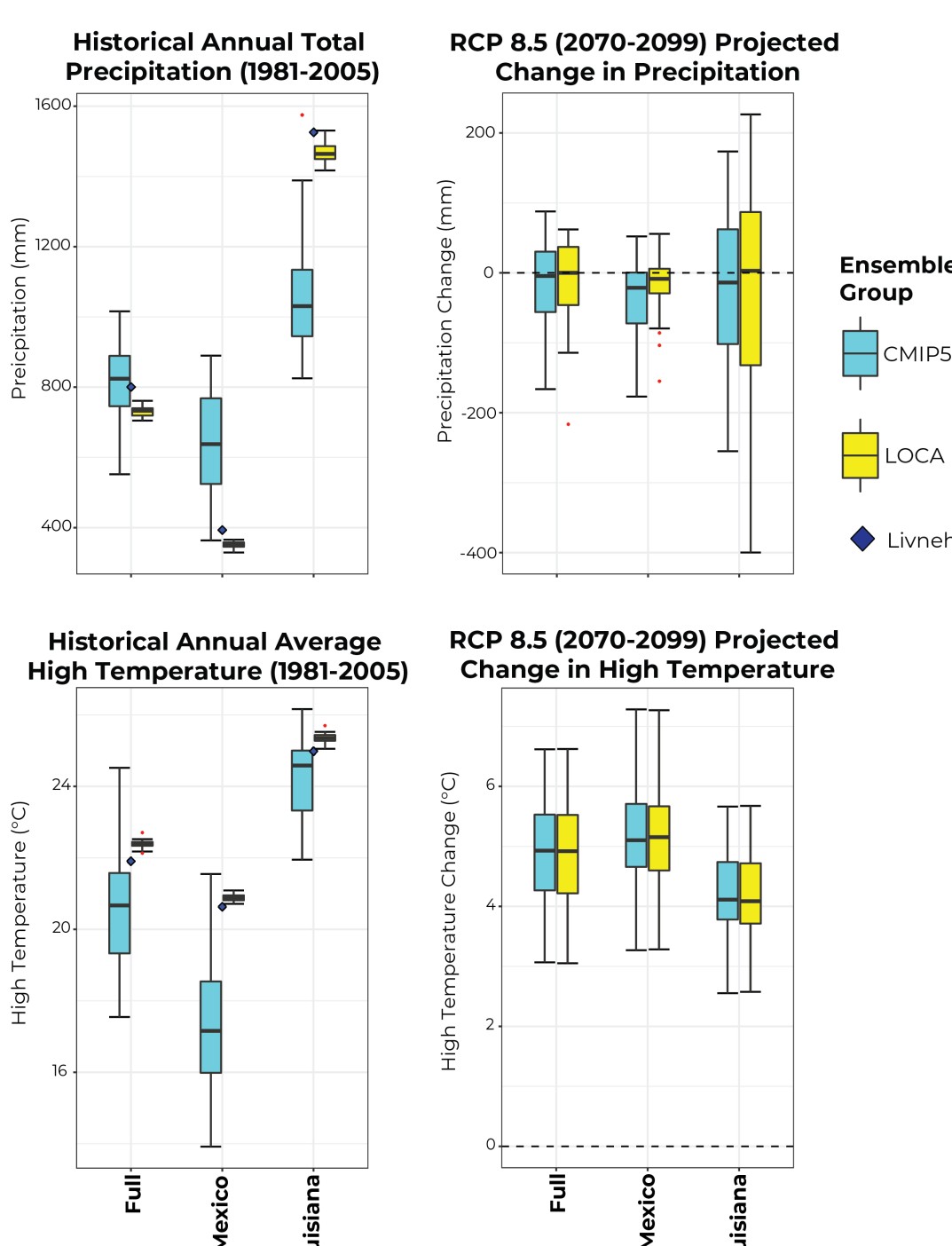

**Figure 5: The unweighted model values across each of the three domains. The left column is during the historical period (1981-2005) and the raw ensemble is compared to the same values from the Livneh observations. The right column is the 2070-2099 projected changes under RCP 8.5 from both ensembles. The top row is for precipitation, the bottom row is for high temperature.**


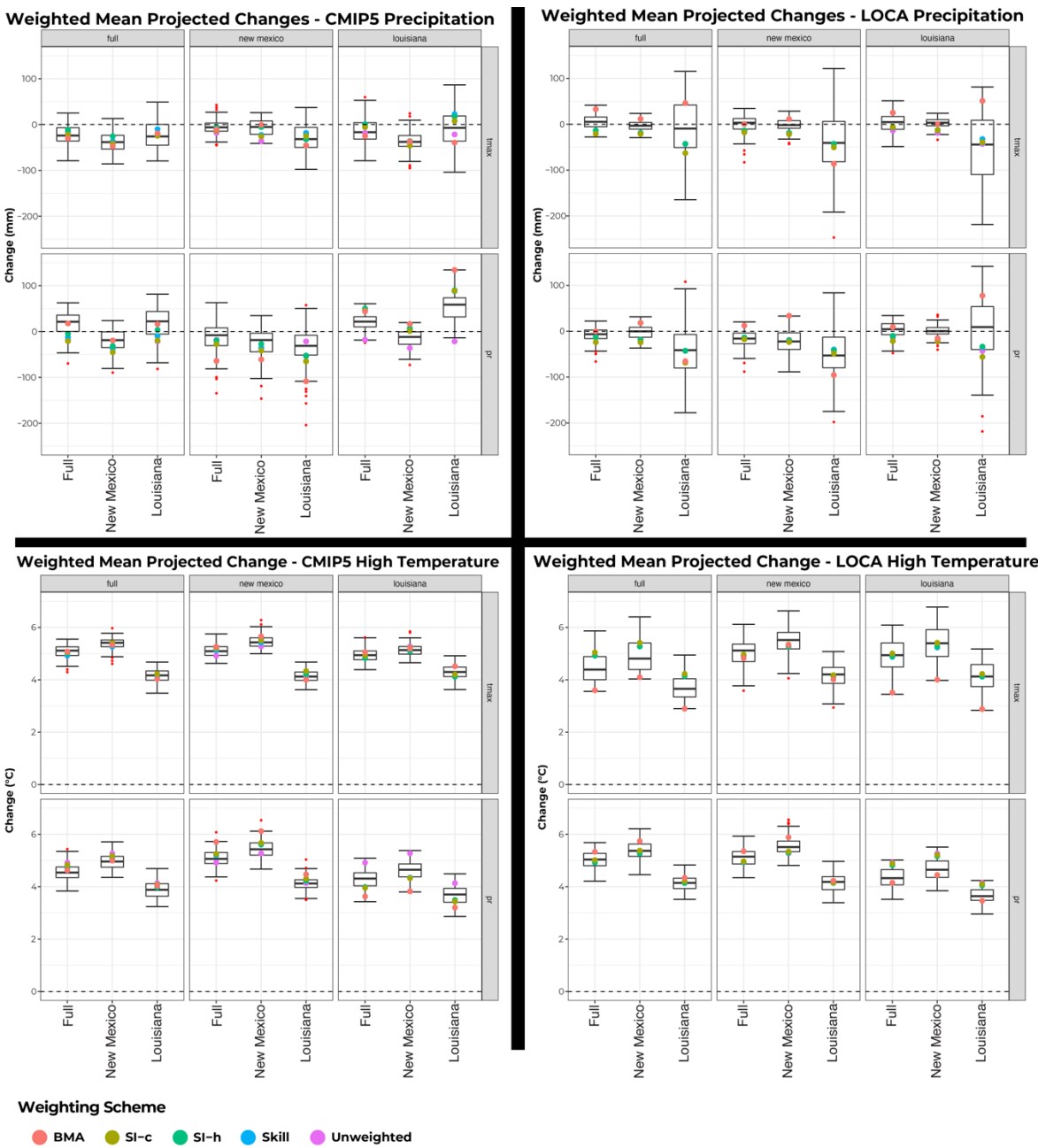

**Figure 6: Mean projected changes in temperature and precipitation using all 48 weighting schemes, applied to all three domains and both variables (tmax and pr). The top group focuses on pr, the bottom row focuses on tmax, the left group focuses on the CMIP5 ensemble, and the right group focuses on the LOCA ensemble. In an individual group, the top row is the results from weighting schemes derived with tmax, and the bottom row is the results from weighting schemes derived with pr. In addition, within an individual group, the left column is the results for weighting derived using the full domain, the middle column is the results for weighting derived using the New Mexico domain, and the right column is the results for weighting derived using the Louisiana domain. Within a given domain and variable, the results are shown from left to right for the domain the weights are applied to. The boxplots are the results from the 100 BMA posterior weights.**


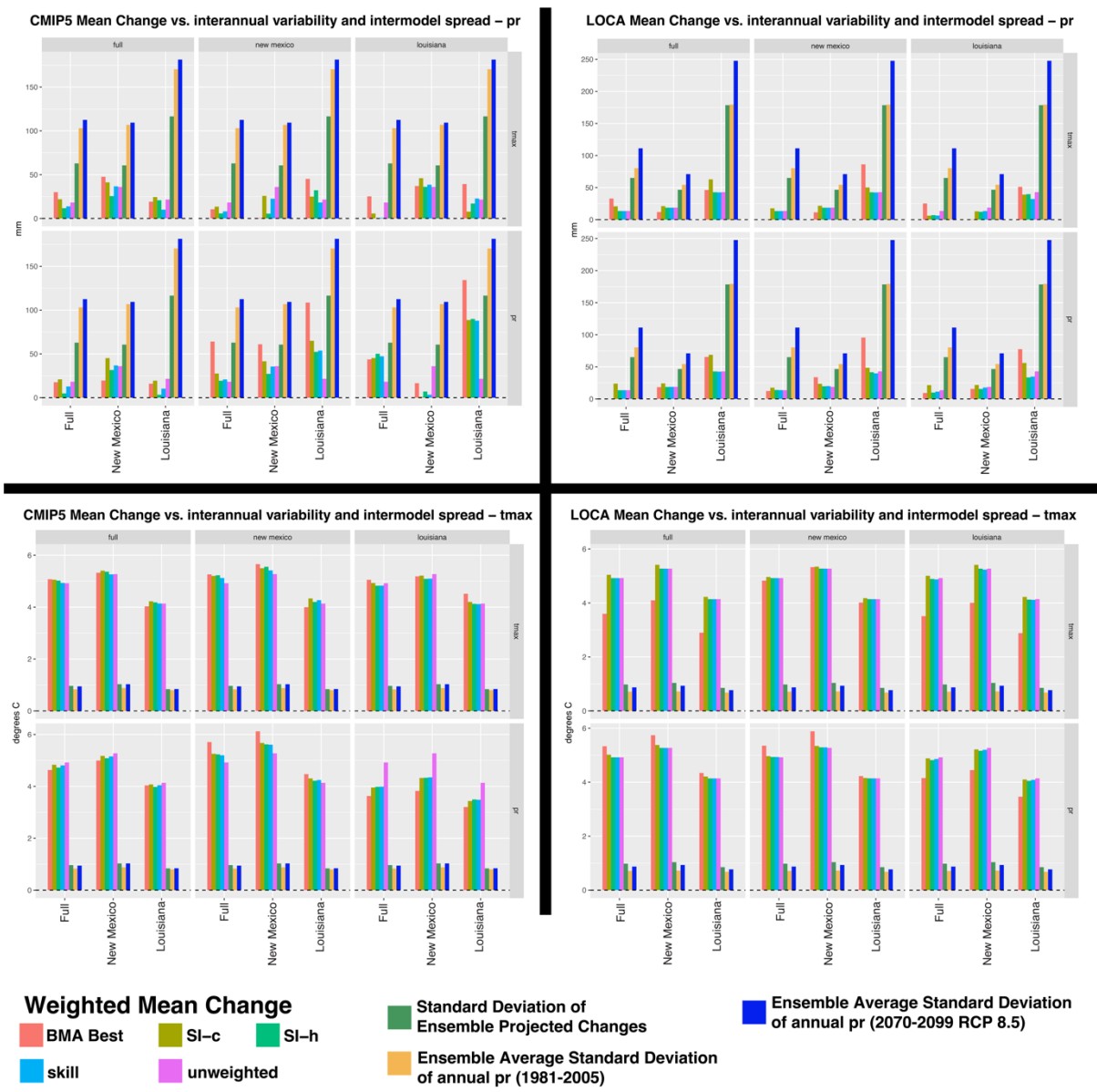

**Figure 7: Absolute value of mean projected changes in temperature and precipitation using all 48 weighting schemes, applied to all three domains and both variables (tmax and pr), the standard deviation of the projected changes from the CMIP5 and LOCA ensembles for both variables, and ensemble average standard deviation of annual precipitation and temperature for both the historical and future periods (no weighting is used to calculate any standard deviations). The top group focuses on pr, the bottom row focuses on tmax, the left group focuses on the CMIP5 ensemble, and the right group focuses on the LOCA ensemble. In an individual group, the top row is the results from weighting schemes derived with tmax, and the bottom row is the results from weighting schemes derived with pr. In addition, within an individual group, the left column is the results for weighting derived using the full domain, the middle column is the results for weighting derived using the New Mexico domain, and the right column is the results for weighting derived using the Louisiana domain. Within a given domain and variable, the results are shown from left to right for the domain the weights are applied to.**

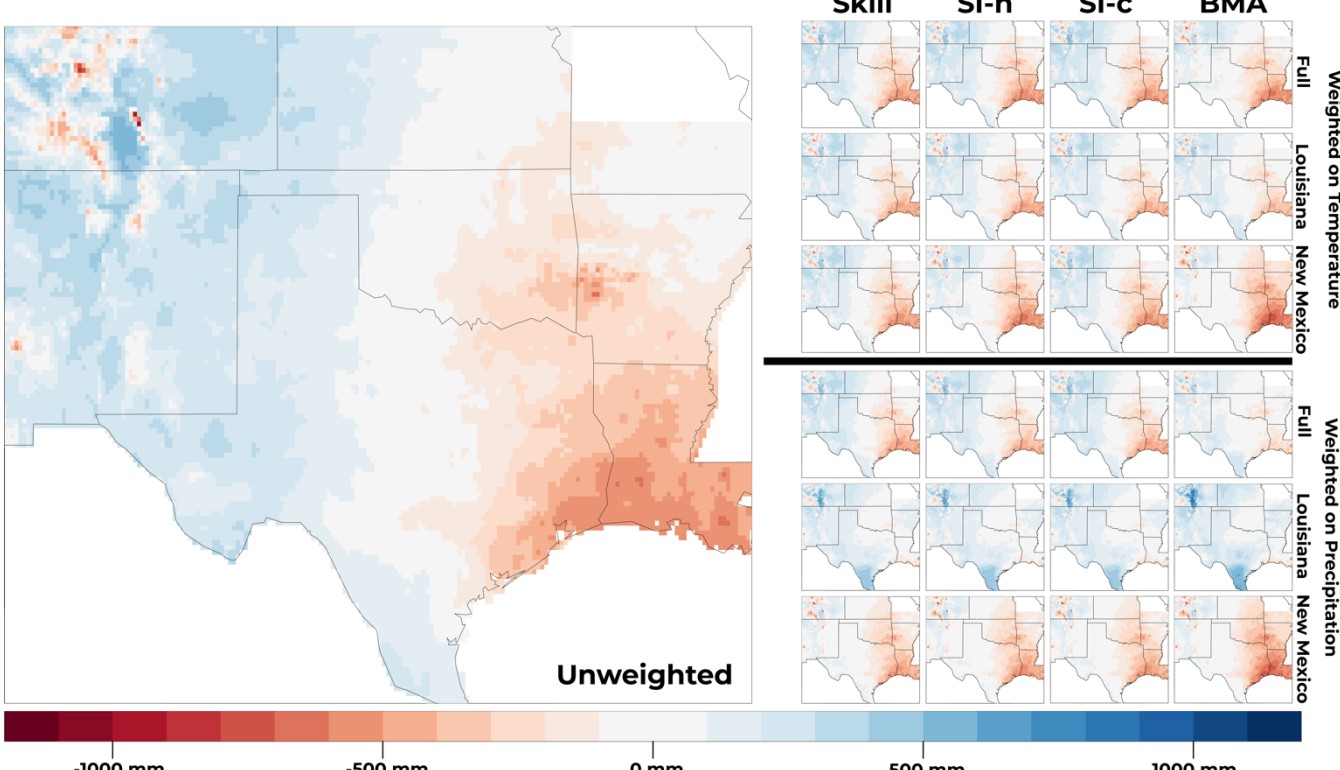

**Figure 8: Bias of CMIP5 ensemble mean precipitation (1981-2005) from the unweighted ensemble (left) and each weighted ensemble mean (right). On the right side, the columns from left to right are for the Skill, SI-h, SI-c, and BMA weighting schemes respectively. On the right side, the top group of twelve plots are the results for weights derived using temperature (tmax) and the bottom group of twelve plots are the results for weights derived using precipitation (pr). Within a group of twelve on the right-hand side, the top row is for weights deriving using the full domain, the middle row is for weights derived using the Louisiana domain, and the bottom row is for weights derived using the New Mexico domain.**


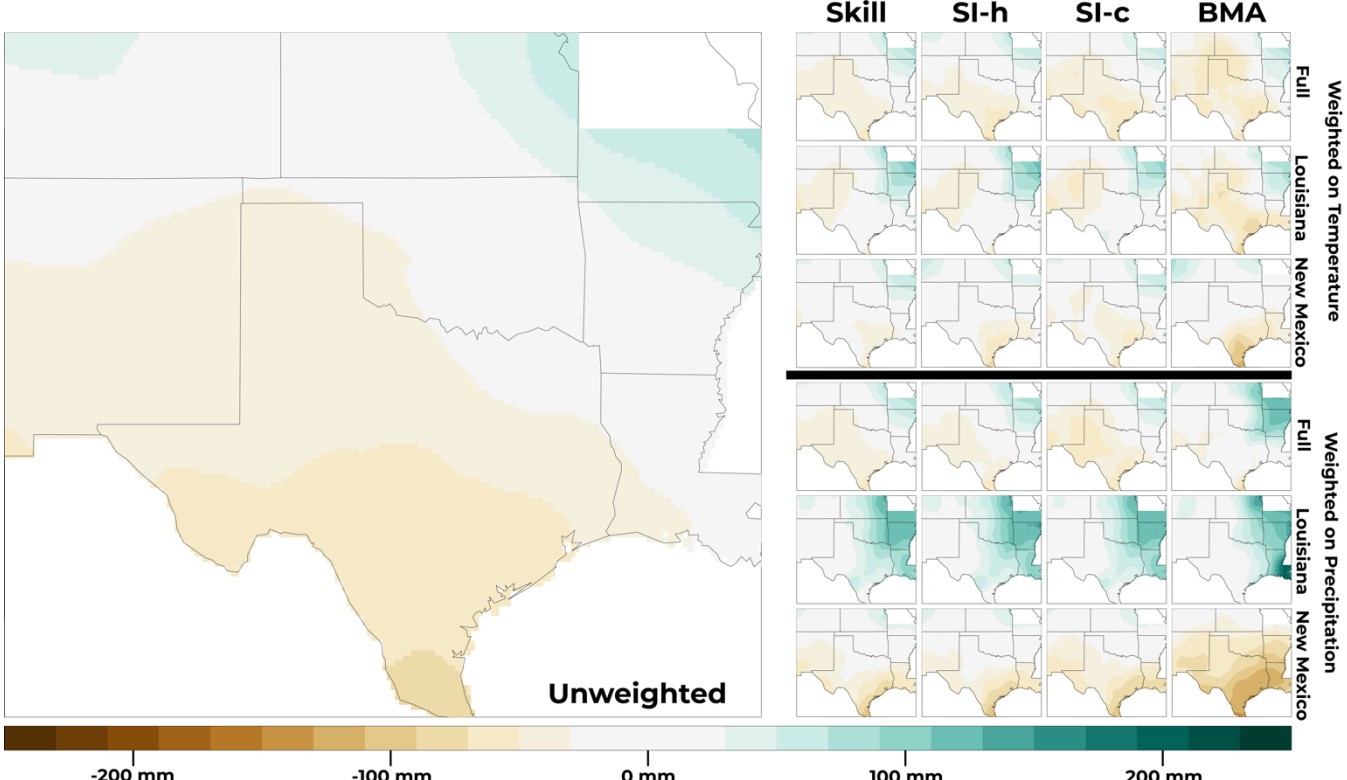

**Figure 9: CMIP5 ensemble mean projected precipitation change (2070-2099, RCP 8.5) from the unweighted ensemble (left) and**
**each weighted ensemble mean (right). On the right side, the columns from left to right are for the Skill, SI-h, SI-c, and BMA weighting schemes respectively. On the right side, the top group of twelve plots are the results for weights derived using temperature (tmax) and the bottom group of twelve plots are the results for weights derived using precipitation (pr). Within a group of twelve on the right-hand side, the top row is for weights deriving using the full domain, the middle row is for weights derived using the Louisiana domain, and the bottom row is for weights derived using the New Mexico domain.**

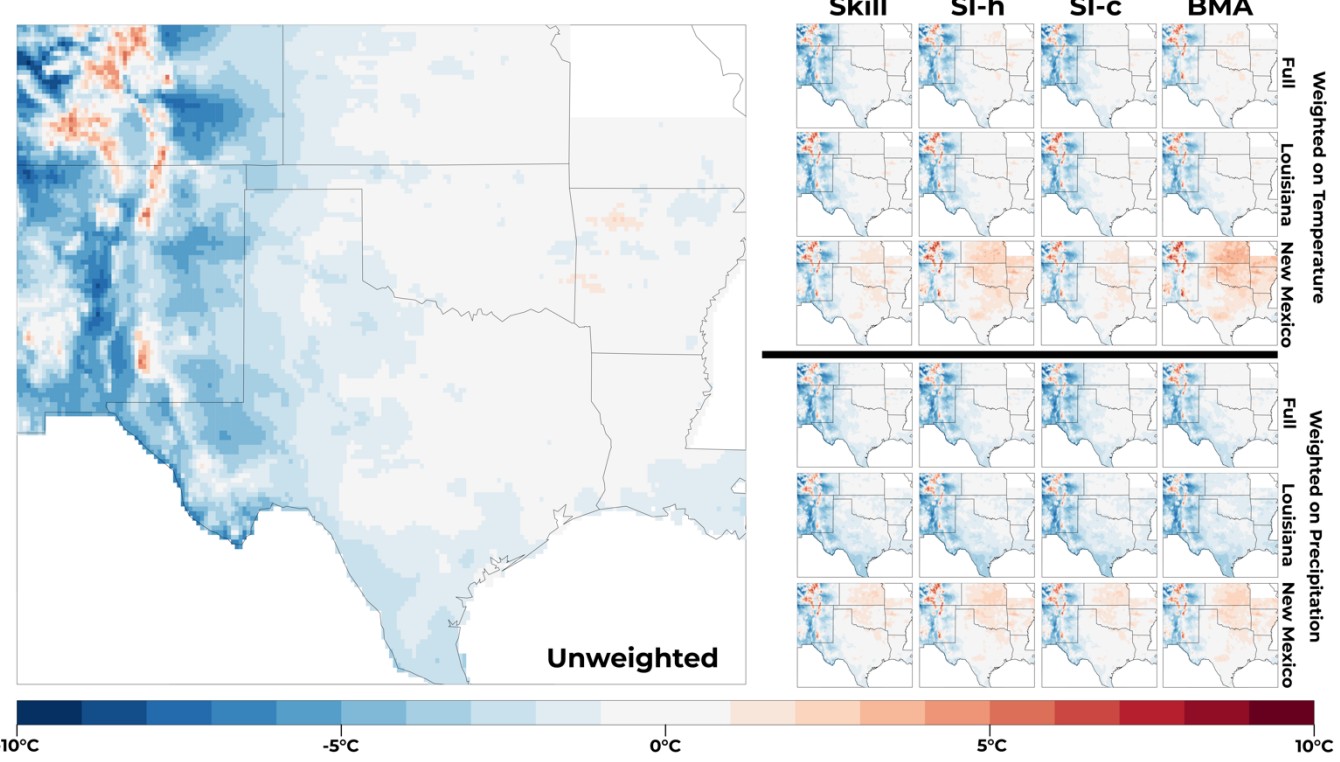


Figure 10: Same as Figure 8, but for the bias of high temperature of the CMIP5 ensemble.

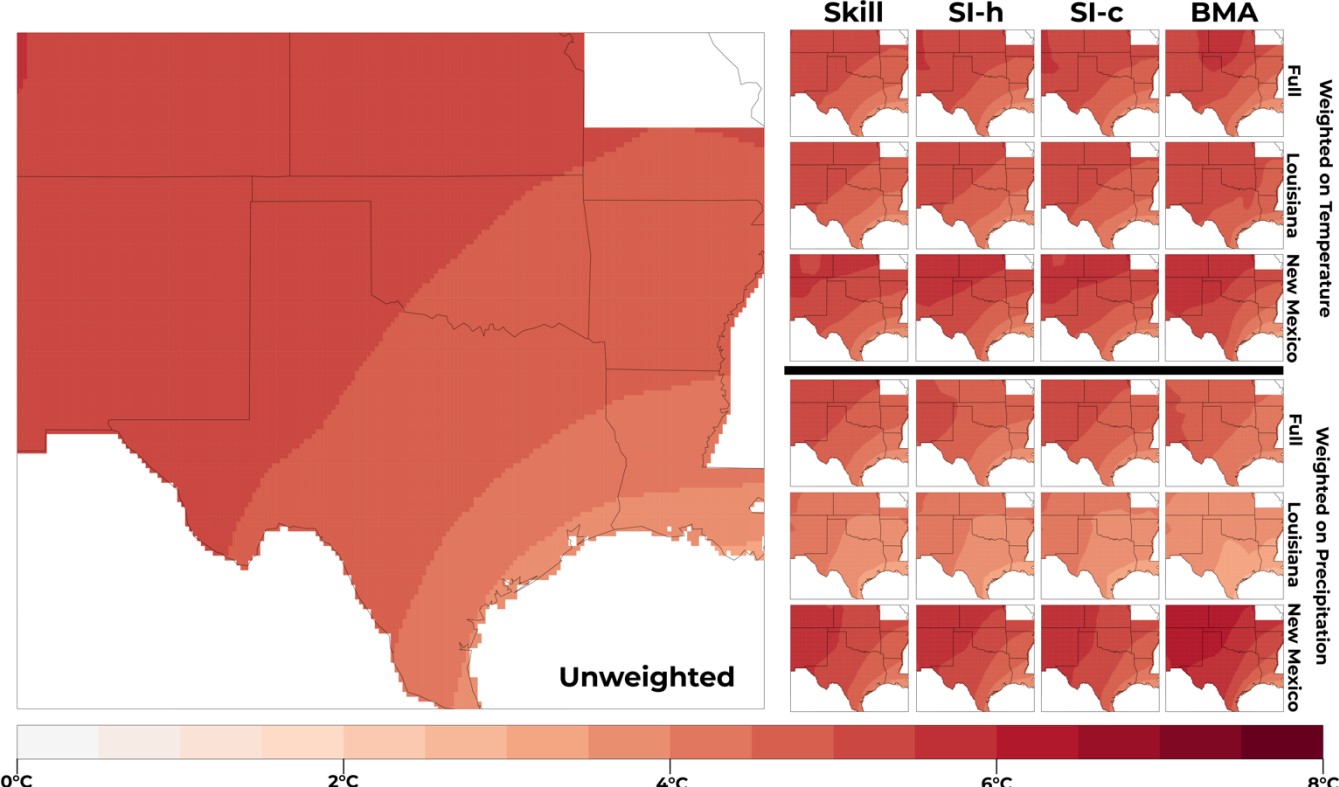

**Figure 11: Same as Figure 9, but for the mean projected change of high temperature from the CMIP5 ensemble.**


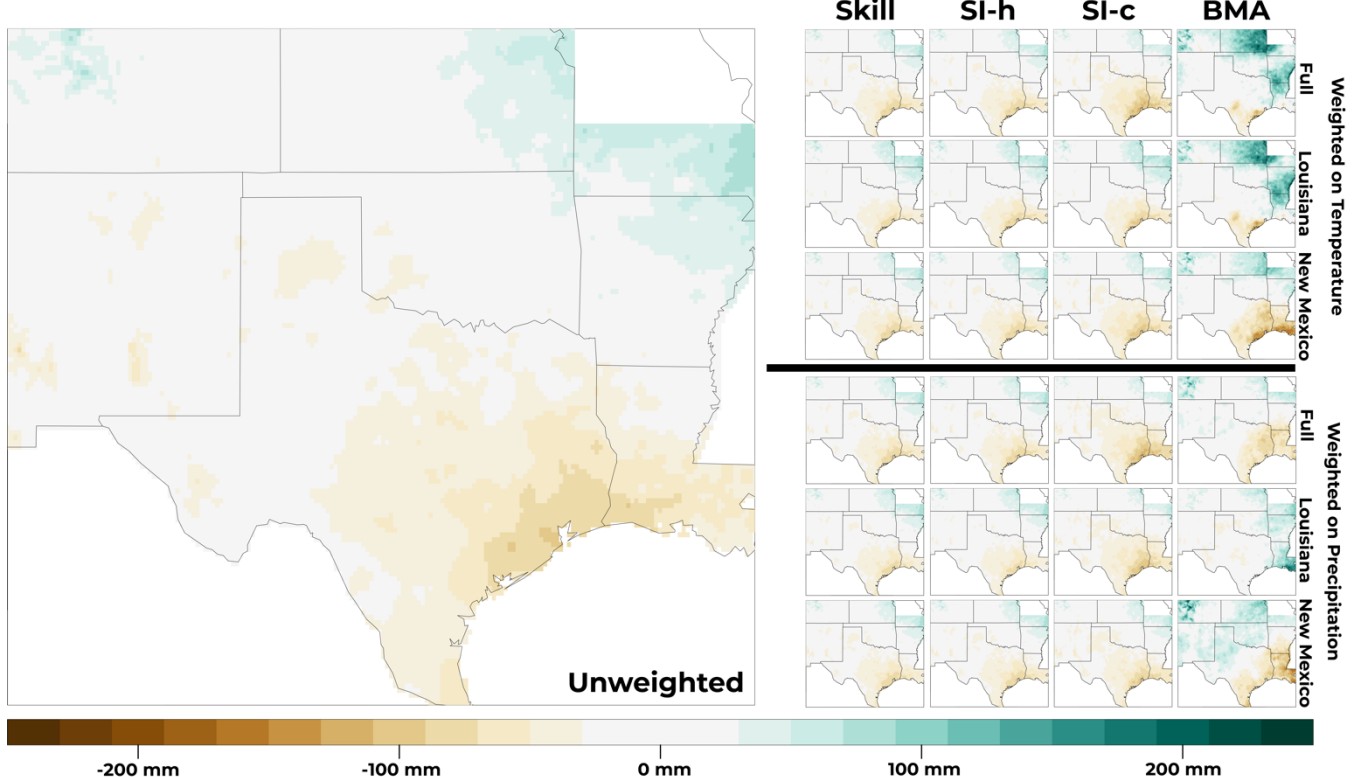

**Figure 12: Same as Figure 9, but for the mean projected change of precipitation from the LOCA ensemble.**

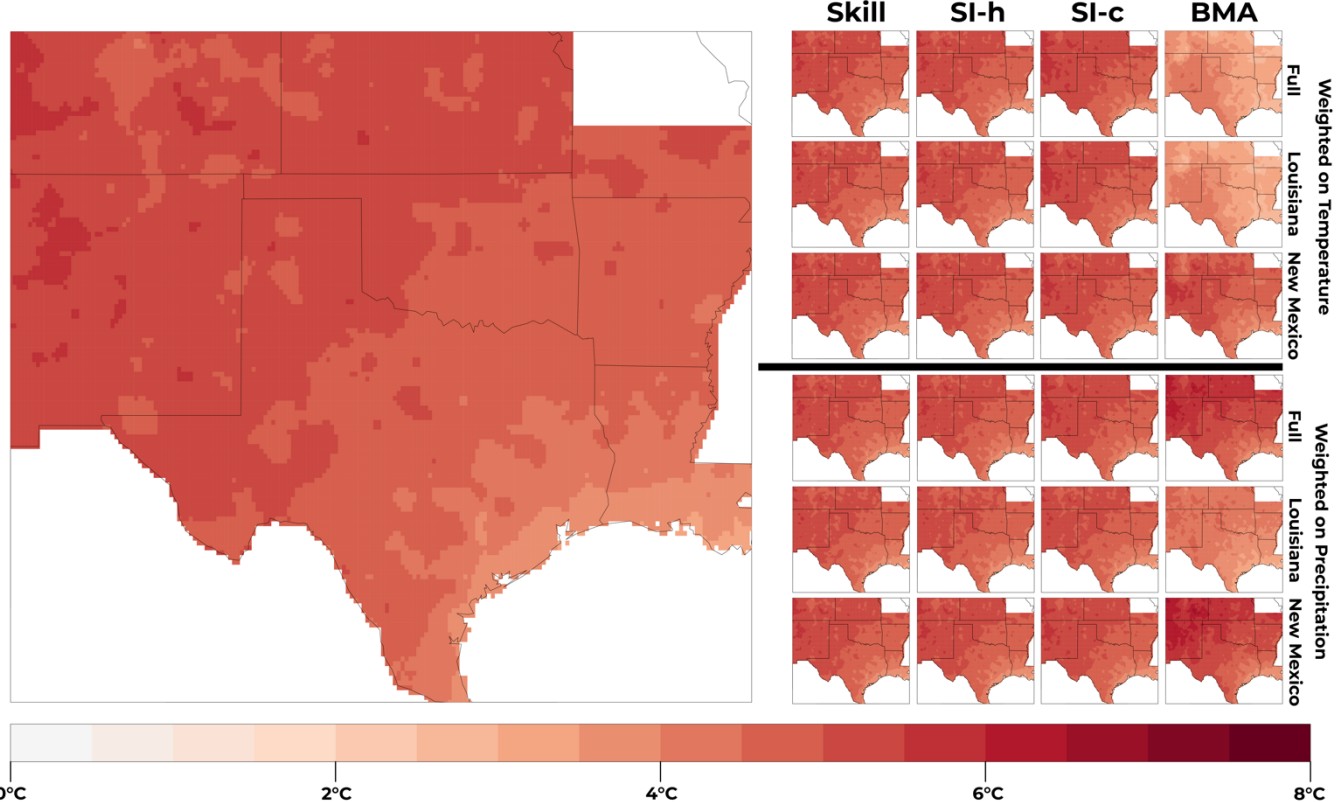

**Figure 13:** Same as Figure 11, but for the mean projected change of high temperature from the LOCA ensemble.






**Table 1: Top three highest weighted models from each of the 48 weighting combinations.**

| Domain Weighting is Based On | Variable Weighting is Based On | Ensemble | Skill | SI-h | SI-c | BMA |
|---|---|---|---|---|---|---|
| Full | tmax | CMIP5 | ACCESS1-0 | CanESM2 | CSIRO-Mk3-6-0 | CSIRO-Mk3-6-0 |
| | | | CSIRO-Mk3-6-0 | CSIRO-Mk3-6-0 | ACCESS1-0 | MPI-ESM-MR |
| | | | CMCC-CMS | MIROC-ESM | CMCC-CM | CMCC-CM |
| | | LOCA | MRI-CGCM3 | MRI-CGCM3 | MRI-CGCM3 | MRI-CGCM3 |
| | | | MIROC-ESM | MIROC-ESM | GISS-E2-R | CanESM2 |
| | | | CESM1-BGC | CESM1-BGC | IPSL-CM5A-MR | FGOALS-g2 |
| | pr | CMIP5 | EC-EARTH | ACCESS1-3 | CMCC-CM | ACCESS1-3 |
| | | | CMCC-CM | EC-EARTH | ACCESS1-0 | EC-EARTH |
| | | | ACCESS1-0 | GISS-E2-R | EC-EARTH | CMCC-CM |
| | | LOCA | CESM1-BGC | CanESM2 | IPSL-CM5A-MR | MIROC-ESM |
| | | | CanESM2 | MIROC-ESM | ACCESS1-0 | CanESM2 |
| | | | MIROC-ESM | CESM1-BGC | CMCC-CM | CESM1-BGC |
| Louisiana | tmax | CMIP5 | ACCESS1-3 | ACCESS1-3 | ACCESS1-3 | CMCC-CM |
| | | | CMCC-CMS | MPI-ESM-MR | ACCESS1-0 | ACCESS1-3 |
| | | | MPI-ESM-LR | CMCC-CMS | MPI-ESM-LR | MIROC5 |
| | | LOCA | MRI-CGCM3 | MIROC-ESM | MIROC-ESM-CHEM | MRI-CGCM3 |
| | | | MIROC-ESM | MRI-CGCM3 | MRI-CGCM3 | GISS-E2-H |
| | | | ACCESS1-3 | ACCESS1-3 | GFDL-CM3 | GFDL-ESM2M |
| | pr | CMIP5 | ACCESS1-3 | ACCESS1-3 | ACCESS1-3 | GISS-E2-R |
| | | | GISS-E2-R | GISS-E2-R | GISS-E2-R | ACCESS1-3 |
| | | | EC-EARTH | EC-EARTH | EC-EARTH | MIROC-ESM-CHEM |
| | | LOCA | CCSM4 | GISS-E2-R | GISS-E2-R | CCSM4 |
| | | | GISS-E2-R | CanESM2 | IPSL-CM5A-MR | GISS-E2-R |
| | | | GFDL-ESM2M | CCSM4 | FGOALS-g2 | EC-EARTH |
| New Mexico | tmax | CMIP5 | CanESM2 | CanESM2 | CSIRO-Mk3-6-0 | CanESM2 |

| | | | | | | |
|---|---|---|---|---|---|---|
| | | | CSIRO-Mk3-6-0 | CSIRO-Mk3-6-0 | ACCESS1-0 | CSIRO-Mk3-6-0 |
| | | | ACCESS1-0 | ACCESS1-0 | CanESM2 | IPSL-CM5A-LR |
| | | LOCA | MRI-CGCM3 | MIROC-ESM | MRI-CGCM3 | MRI-CGCM3 |
| | | | MIROC-ESM | MRI-CGCM3 | MIROC-ESM | MIROC-ESM |
| | | | GISS-E2-H | CanESM2 | GFDL-CM3 | FGOALS-g2 |
| | | CMIP5 | CanESM2 | CanESM2 | IPSL-CM5A-MR | CanESM2 |
| | | | IPSL-CM5A-MR | CSIRO-Mk3-6-0 | CanESM2 | IPSL-CM5A-MR |
| | pr | | ACCESS1-3 | IPSL-CM5A-MR | ACCESS1-3 | CSIRO-Mk3-6-0 |
| | | LOCA | MPI-ESM-LR | MPI-ESM-LR | CanESM2 | CanESM2 |
| | | | CanESM2 | CanESM2 | MPI-ESM-LR | MIROC-ESM |
| | | | MIROC-ESM | MIROC-ESM | CMCC-CM | EC-EARTH |







**Table 2: RMSE (mm) of daily precipitation for each weighting strategy applied in all three domains. Weighting strategies include the weighting scheme and the ensemble, variable, and domain used to derive model weights.**

| Weighting Strategy | | | | Applied Domain | | |
|---|---|---|---|---|---|---|
| Ensemble | Variable | Domain | Weighting Scheme | Full | New Mexico | Louisiana |
| CMIP5 | pr | Full | Skill | 212.10 | 232.01 | 399.03 |
| | | | SI-h | 196.04 | 217.24 | 346.87 |
| | | | SI-c | 209.86 | 231.81 | 387.89 |
| | | | BMA | 138.18 | 144.07 | 197.90 |
| | | New Mexico | Skill | 188.21 | 403.07 | 109.52 |
| | | | SI-h | 198.05 | 106.89 | 436.66 |
| | | | SI-c | 192.07 | 109.17 | 412.74 |
| | | | BMA | 262.49 | 109.67 | 573.99 |
| | | Louisiana | Skill | 179.65 | 182.57 | 91.68 |
| | | | SI-h | 177.40 | 175.26 | 89.30 |
| | | | SI-c | 180.19 | 186.10 | 92.42 |
| | | | BMA | 220.92 | 212.44 | 109.47 |
| | tmax | Full | Skill | 223.77 | 240.32 | 431.97 |
| | | | SI-h | 226.69 | 225.97 | 458.80 |
| | | | SI-c | 222.72 | 233.96 | 431.94 |
| | | | BMA | 205.24 | 136.17 | 477.53 |
| | | New Mexico | Skill | 212.83 | 438.97 | 197.46 |
| | | | SI-h | 219.89 | 145.69 | 498.12 |
| | | | SI-c | 219.94 | 207.25 | 450.48 |
| | | | BMA | 255.59 | 109.41 | 586.54 |
| | | Louisiana | Skill | 196.94 | 211.71 | 373.35 |
| | | | SI-h | 191.86 | 195.90 | 381.33 |
| | | | SI-c | 180.19 | 193.73 | 351.64 |
| | | | BMA | 167.19 | 171.51 | 313.55 |
| LOCA | pr | Full | Skill | 66.67 | 79.58 | 60.58 |
| | | | SI-h | 66.66 | 60.59 | 79.54 |
| | | | SI-c | 66.81 | 60.61 | 80.10 |
| | | | BMA | 55.87 | 55.02 | 69.31 |
| | | New Mexico | Skill | 66.26 | 59.94 | 79.54 |
| | | | SI-h | 66.24 | 60.04 | 79.37 |
| | | | SI-c | 66.53 | 59.60 | 80.56 |
| | | | BMA | 56.76 | 53.94 | 62.77 |

| | | | | | |
|---|---|---|---|---|---|
| | Louisiana | Skill | 65.72 | 60.43 | 76.29 |
| | | SI-h | 65.49 | 60.66 | 75.04 |
| | | SI-c | 66.15 | 61.31 | 75.34 |
| | | BMA | 61.77 | 60.30 | 55.27 |
| tmax | Full | Skill | 66.77 | 60.63 | 79.71 |
| | | SI-h | 66.77 | 60.63 | 79.74 |
| | | SI-c | 67.32 | 60.98 | 80.82 |
| | | BMA | 66.78 | 62.76 | 80.40 |
| | New Mexico | Skill | 66.77 | 60.62 | 79.72 |
| | | SI-h | 66.77 | 60.62 | 79.74 |
| | | SI-c | 66.94 | 60.76 | 80.59 |
| | | BMA | 63.02 | 59.10 | 81.92 |
| | Louisiana | Skill | 66.69 | 60.89 | 78.65 |
| | | SI-h | 66.54 | 60.74 | 79.56 |
| | | SI-c | 68.39 | 61.41 | 81.79 |
| | | BMA | 69.53 | 65.02 | 83.67 |







**Table 3: RMSE (°C) of daily precipitation for each weighting strategy applied in all three domains. Weighting strategies include the weighting scheme and the ensemble, variable, and domain used to derive model weights.**

| | Weighting Strategy | | | Applied Domain | | |
|---|---|---|---|---|---|---|
| Ensemble | Variable | Domain | Weighting Scheme | Full | New Mexico | Louisiana |
| CMIP5 | pr | Full | Skill | 2.19 | 3.67 | 0.86 |
| | | | SI-h | 2.19 | 3.64 | 0.93 |
| | | | SI-c | 2.29 | 3.80 | 0.93 |
| | | | BMA | 2.04 | 3.37 | 0.80 |
| | | New Mexico | Skill | 1.79 | 2.78 | 0.71 |
| | | | SI-h | 1.76 | 2.48 | 0.75 |
| | | | SI-c | 1.81 | 2.86 | 0.71 |
| | | | BMA | 1.83 | 2.58 | 0.81 |
| | | Louisiana | Skill | 2.29 | 3.61 | 1.36 |
| | | | SI-h | 2.27 | 3.57 | 1.34 |
| | | | SI-c | 2.33 | 3.66 | 1.42 |
| | | | BMA | 2.47 | 3.75 | 1.79 |
| | tmax | Full | Skill | 2.09 | 3.50 | 0.70 |
| | | | SI-h | 1.96 | 3.24 | 0.71 |
| | | | SI-c | 2.09 | 3.48 | 0.74 |
| | | | BMA | 1.58 | 2.51 | 0.61 |
| | | New Mexico | Skill | 1.77 | 2.72 | 0.66 |
| | | | SI-h | 1.87 | 2.21 | 0.92 |
| | | | SI-c | 1.78 | 2.77 | 0.66 |
| | | | BMA | 2.10 | 2.07 | 1.08 |
| | | Louisiana | Skill | 1.92 | 3.21 | 0.49 |
| | | | SI-h | 1.90 | 3.19 | 0.48 |
| | | | SI-c | 1.91 | 3.17 | 0.49 |
| | | | BMA | 1.86 | 3.15 | 0.40 |
| LOCA | pr | Full | Skill | 0.43 | 0.49 | 0.40 |
| | | | SI-h | 0.43 | 0.49 | 0.40 |
| | | | SI-c | 0.44 | 0.49 | 0.41 |
| | | | BMA | 0.34 | 0.44 | 0.33 |
| | | New Mexico | Skill | 0.43 | 0.49 | 0.40 |
| | | | SI-h | 0.43 | 0.49 | 0.40 |
| | | | SI-c | 0.44 | 0.49 | 0.40 |
| | | | BMA | 0.36 | 0.44 | 0.37 |

| | | | | | |
|---|---|---|---|---|---|
| | Louisiana | Skill | 0.43 | 0.49 | 0.39 |
| | | SI-h | 0.43 | 0.49 | 0.39 |
| | | SI-c | 0.43 | 0.49 | 0.39 |
| | | BMA | 0.42 | 0.49 | 0.35 |
| | Full | Skill | 0.43 | 0.49 | 0.40 |
| | | SI-h | 0.43 | 0.49 | 0.40 |
| | | SI-c | 0.43 | 0.49 | 0.39 |
| | | BMA | 0.28 | 0.43 | 0.21 |
| tmax | New Mexico | Skill | 0.43 | 0.49 | 0.40 |
| | | SI-h | 0.43 | 0.49 | 0.40 |
| | | SI-c | 0.43 | 0.49 | 0.39 |
| | | BMA | 0.31 | 0.43 | 0.21 |
| | Louisiana | Skill | 0.42 | 0.48 | 0.37 |
| | | SI-h | 0.41 | 0.48 | 0.36 |
| | | SI-c | 0.42 | 0.48 | 0.35 |
| | | BMA | 0.29 | 0.44 | 0.21 |


**Table 4: RMSE of precipitation (mm) and high temperature (°C) for unweighted ensembles.**

| | | Domain | | |
|---|---|---|---|---|
| **Variable** | **Ensemble** | **Full** | **New Mexico** | **Louisiana** |
| pr | CMIP5 | 239.25 | 266.19 | 449.79 |
| | LOCA | 66.79 | 60.63 | 79.76 |
| tmax | CMIP5 | 2.39 | 3.95 | 0.96 |
| | LOCA | 0.43 | 0.49 | 0.40 |
