# Peer review of "Assessing sensitivities of climate model weighting to multiple methods, variables, and domains in the south-central United States"

_Earth System Dynamics, 2022_

## Author Comment (AC1)

CMIP5 Mean Change vs. interannual variability and intermodel spread – pr

---

## Author Response (AR1)

Reviewer 1 comments

In this study, the authors set up a systematic exploration of several combinations of choices in multimodel ensemble weighting schemes, and describe the resulting projections when weighting CMIP5 models and their downscaled and bias-corrected LOCA versions.

*We thank the reviewer for taking the time to thoroughly review our manuscript. Our responses to the comments provided are in italics at points in the reviewer's comments.*

The authors offer that the value of this work is in this systematic exploration of the effects of weighting, but I am sorry to say that, aside from some very nice and thoughtful discussion of general issues (which by the way have been treated in some depth by a guidance document for the IPCC AR5 report as early as 2010, available here https://www.wcrp-climate.org/wgcm/references/IPCC_EM_MME_GoodPracticeGuidancePaper.pdf, and more recently in a review paper by Abramowitz et al. (2019) https://doi.org/10.5194/esd-10-91-2019), and the appreciation of the large amount of work that the authors have undertaken, I come away from this study only reinforcing what we all already knew: that different weighting schemes produce different results and nobody knows how to interpret the real value of those differences and what to do about it.

*We respectfully disagree with the assessment of the reviewer. The debate over climate model weighting is precisely why we chose to invest the time and energy into this extensive study. Nobody knows what to do about the differences in results between different methods and applications of climate model weighting. So, we underwent this extensive and comprehensive research matrix of results and answered some of these outstanding questions in the community. In addition, several authors are involved in producing the Fifth National Climate Assessment (NCA) in the United States. The authors of this study are all directly and indirectly involved with the NCA discussions, led by the United States Global Change Research Program (USGCRP), surrounding downscaling and multi-model ensemble weighting for the Fifth NCA. There are 10-15 people representing multiple agencies of the United States government discussing issues associated with downscaling and model weighting during bi-weekly meetings over the past two years. The effort in this study, and the questions of interest in this study, delivers research on questions of interest to the USGCRP and the broader discussion group. No other study has comprehensively answered these many questions in one study, such as applying model weighting based on the climate variables of interest, the domain of interest, different model weighting strategies, or the dataset used (GCM or downscaled).*

*While IPCC AR5 report from 2010 provides general guidance, it does not include analysis or investigation into the recommendations provided, whereas our study does. For example, in Section 3.5 of the IPCC AR5 report, the authors discuss recommendations for regional assessments, and conclude that "Particular climate projections should be assessed against the broader context of multiple sources (e.g., regional climate models, statistical downscaling) of regional information on climate change (including multi-model global simulations), recognizing that real and apparent contradictions may exist between information sources which need physical understanding." This is precisely what our study aims to do, by utilizing data from both global models as well as their downscaled counterparts. This is just one example of how the*

*IPCC AR5 report makes recommendations and does not perform any investigations, where our study does apply the research needed to address such recommendation. In addition, our study does so and goes on to make several recommendations on the appropriate use of multi-model ensemble weighting, which is summarized in the abstract and conclusions sections of our study.*

*While Abramowitz et al. (2019) covers the concept of model dependence, our manuscript goes much further. Through the various weighting schemes, this manuscript covers different approaches to dealing with model independence. Two of the weighting schemes account for model dependence using the method created by Sanderson et al (2017) that accounts for model dependence in the historical simulation, and a variation of the Sanderson et al. (2017) method that accounts for model dependence in the future climate change signal. The former method has been used in previous studies, but the latter method is not a common approach to dealing with model independence and is not covered in the paper by Abramowitz et al. 2019. In addition, this study also includes a Bayesian Model Averaging (BMA) weighting scheme that approaches the model dependence problem over multiple moments of the distribution. Bayesian approaches are only mentioned in passing by Abramowitz et al. 2019.*

*Given the above, we believe that our study is timely given the debate in the community regarding the use of multi-model weighting.*

In my view, there would be two ways to make this exploration more useful.

First, perform this exercise with a clear accounting of internal and inter-model variability. I don't know what to make of pictures that show me multimodel means and how they differ from one another. The question is, do they differ in a way that is significant, compared to internal variability? And do they differ in a way that is significant with respect to a measure of uncertainty around the multimodel mean, which could be taken (likely underestimating it and therefore possibly favoring the detection of significant differences, but that could be expressed as a caveat) as its standard deviation, computed by the inter-model standard deviation divided by the square root of the ensemble size (at each grid point)?

*We agree that this is a useful additional component and thank the reviewer for pointing out that we should address it, at least as a caveat, in our study. There are two points to be made on this issue raised by the reviewer. First, Figure 5 in our manuscript shows the inter-model variability for the projected changes of both variables for both ensembles and all domains and Figure 6 shows the resulting ensemble means from the various weighting schemes. These two figures clearly indicate that the differences between the means are not as large as the inter-model variability. Second, while these figures do offer some suggestion, we agree that this is not a robust and explicit treatment and does not include a discussion of internal variability. While a full treatment and discussion of the weighting compared to internal and inter-model variability is beyond the scope of this paper, the revised manuscript includes an approximate analysis and an additional figure will be placed after Figures 5-6 with additional discussion.*

*The new Figure 7 (a portion of Figure 7 is included below) includes the absolute value of the ensemble mean changes in comparison with the standard deviation of the projected changes of the ensemble (representing the ensemble mean change), the average of the standard deviation of the annual values of each ensemble member from the historical period, and the average of the standard deviation of the annual values of each ensemble member from the future period. These latter two items have been used as a rough approximation of the internal variability in multiple studies previously (e.g., Hawkins and Sutton, 2009; 2011). As such, the new Figure 7 is of a similar arrangement to Figure 6, but a component of the final figure is provided below. This component is for the precipitation projections from the CMIP5 ensemble, weighted based on tasmax in the top row and weighted based on precipitation in bottom row, and weighted based on the full (left), New Mexico (middle), and Louisiana (right) domains. Within a single plot of bars, the weighted means are for the weighting schemes applied to the full domain (left), New Mexico (middle), and Louisiana (right). For a single group of bars, the multi-model means from the various schemes are plotted directly against our proxies for the inter-model and internal variability. From this sample we can say that in most cases, the differences between ensemble means are not larger than the inter-model or internal variability. However, there are some cases where the differences between these means are comparable to or larger than inter-model variability or comparable to the internal variability. For example, for ensemble means weighted for Louisiana precipitation and applied to Louisiana precipitation (circled in red below for reference), the difference between the BMA ensemble mean and the unweighted mean is comparable to the inter-model variability. As a further example, the difference between BMA ensemble mean created based on Louisiana precipitation and all the weighted ensemble means created based on full domain precipitation (circled in black below for reference) is also comparable to inter-model variability and internal variability. The comments discussed here are included and expanded upon in section 3.3 of the revised manuscript.*

[Figure]

***Component of New Figure 7.*** *Absolute value of mean projected changes in precipitation from the CMIP5 ensemble using multi-model weights produced with all four weighting schemes applied to all three domains and both variables (tmax and pr) plotted alongside the standard deviation of the CMIP5 ensemble and the ensemble average standard deviation of annual precipitation for both the historical and future periods (no weighting is used to calculate any standard deviations). In the new Figure 7, which will take a similar form to Figure 6, this would be the top left group of plots. The top row is the results from weighting schemes derived with tmax, and the bottom row is the results from weighting schemes derived with pr. In addition, within an individual group, the left column is the results from weighting derived using the full domain, the middle column is the results for weighting derived using the New Mexico domain, the right column is the results for weighting derived using the Louisiana domain. Within a given domain and variable, the results are shown from left to right for the domain the weights are applied to. For simplicity, the BMA best weights are used (and the boxplots from Figure 6 are omitted). The standard deviations are in all cases from the unweighted ensemble.*

Second, perform a perfect model exercise where one model furnishes the truth, current and future, and the rest of the models undergo this exercise in variation of weights (derived using the left-out model historical portion as observations), so that besides ascertaining that the weights have diverse effects, we can start seeing something about the value of applying them: Do they produce anything more accurate than the unweighted projection? Which of the choices does that better, if any?

The need to take into account internal variability requires the "true model" to be one that has

produced initial condition ensembles, but there are plenty of CMIP5-era large ensembles now available through US CLIVAR SMILEs (https://www.cesm.ucar.edu/projects/community-projects/MMLEA/), and the authors could easily choose one which has also participated in CMIP5 (e.g., CESM1, CanESM, MPI).

*While we agree that with the reviewer that this is a useful exercise, the question of the stationarity of weighting schemes is beyond the scope of this analysis and worthy of a manuscript in and of itself. In addition, the perfect model method assumes that the model that is chosen is a good approximation to the truth and that the future climate simulated in this model, along with the change in climate that occurs within its simulations, is representative of actual climate change and that the other models in the ensemble should have the same change signal. For example, in Brunner et al., (2019, ERL) it mentions "For all regions there is also a chance that the skill decreases due to the weighting. This can happen if the perfect model has a very different response to future forcing compared to the other models, leading to the weighted multi-model ensemble moving further away from the 'truth'." Furthermore, when applying weighting on the LOCA downscaled data, this perfect model test becomes irrelevant since all the models are bias-corrected to apply the downscaling. The author team is considering examining the question in a future study where we would also vary the model used as the absolute truth to examine some of the assumptions associated with the perfect model approach. The following is included in the conclusions of the revised manuscript: "Third, this study does assume stationarity in the multi-model ensemble weights and resulting weighted means. Future research will examine the accuracy and sensitivity using a perfect model exercise (such as what is described by Dixon et al. 2016) to test the stationarity assumption associated with ensemble weighting."*

A study that can tell me something more than "things look different" and can distinguish differences that are simply noise from differences in the signal estimated by these various weighting schemes, then proceed to tell me which one of these weighting schemes, if any, produces projections closest to the "truth" would be really valuable and a real step forward in this old and somewhat frustrating debate.

And I realize that using the perfect model set-up pre-empties the idea of using LOCA, but I would argue that the loss would be more than balanced by the gain in interpretability of the results. Plus, the bias correction of LOCA makes the value of using performance-based weights rather debatable, and my guess is that the differences that surface in that part of the exercise would turn out to be drowned by internal variability if that was accurately accounted for (given that observations used to bias-correct are also just one realization, heavily affected by internal variability at these grid-point scales).

*The authors recognize that the use of statistical downscaling methods may make it debatable to use ensemble weighting. It was in recognition of this debate that we chose to include the ensemble of LOCA downscaled climate models in our experimental matrix, to try and provide answers to this debate.*

I also would like to raise a point about impact modelling. The authors discuss more than once the relevance of the weighting choice for impact modelers, but I would like to be better convinced of that. My experience of impact model(er)s is that they need climate information that looks like reality (one realization of it, or multiple realization of it) not like a big smooth mean. So I agree that the multimodel mean (weighted or unweighted) might be relevant as a synthetic "bird-eye view" of how climate impact-drivers look in the future, and can inform discussions and produce useful catalogs of maps in documents like IPCC or NCA assessments. However, when it comes to impact modeling, my expectation is that feeding multimodel means to a process or empirical model would be nonphysical. Even a large, global scale impact modeling exercise like ISIMIP (https://www.isimip.org/) has provided individual realizations of multiple models for use in its "children" exercises. I would think that using temperature, precipitation and whatever else is needed that behave like reality as input to the impact model, and only after having produced the impact response worrying about averaging, is even more necessary for regional impact assessments like the ones that the authors are mostly concerned about. If I'm wrong, I will happily stand corrected, but in that case I would like to see citations of current impact modeling studies that use multimodel ensemble means.

*What the reviewer describes is precisely why Bayesian Model Averaging (BMA) is utilized. BMA has been proven to be a useful model weighting tool because BMA does not simply provide a single smooth multi-model mean, but instead provides samples from a posterior of model weights in which each sample produces a realistic model average that is not necessarily smoothed out but keeps the internal variability of the system intact. We reported on the mean from the BMA distribution, but in fact the samples can be investigated independently (as in Figure 6 of this study), for both looking at future climate change signals but also for driving impact models. We refer the reviewer to Massoud et al, (2019, 2020a), where BMA was extensively reported on and explained in detail. For example, Massoud et al., 2020a show how climate variability, such as annual and interannual variability of precipitation, is better explained with a BMA model average compared to most individual models, and especially compared to an unweighted model mean, which in comparison washes out any variability in the climate system it represents. In fact, the NCA's 5th assessment report will be utilizing BMA for these reasons, among others, as their chosen tool for applying model averaging. In addition, the study does point to early signs that multi-model ensemble means created with weighting schemes are being used for impact assessments (one of several cited in the study – Skahill et al. 2021 – https://doi.org/10.3390/cli9090140). The text has been revised to make these points clearer and incorporate additional references as suggested by Reviewer 2.*

In conclusion, my assessment of this work is that it represent a very diligent and substantial exercise, informed by thoughtful considerations, but does not help to advance the field until it takes up a better treatment of internal and model variability that could help to determine the significance of the differences resulting from the various weighting schemes, and until it can say something about the usefulness of weighting at all. I tried to suggest ways to do just that. I would be very excited to see the new results, which I hope would not be too difficult to produce, given the efficient machinery that the authors have obviously already in place.

*We thank the reviewer for their comments and critiques. As mentioned above, we have provided an analysis of the ensemble means compared to the internal and inter-model variability (e.g., Figure 7 component shown above in this response letter). We have also adjusted the text to provide some discussion and caveats as mentioned above.*

**Citation**: https://doi.org/10.5194/esd-2022-15-RC1

Reviewer 2 Comments

Review of 'To weight or not to weight: assessing sensitivities of climate model weighting to multiple methods, variables, and domains' by Wootten and colleagues

*We thank the reviewer for taking the time to thoroughly review our manuscript. Our responses to the comments provided are in italics at points in the reviewer's comments.*

The manuscript presents a comparison between what seems to be 2 different climate model weighting schemes in 4 different setups. Weights are based on 2 different variables, 3 different domains, and 2 different model ensembles resulting in 48 different sets of weights. These are applied to the same 2 variables and 3 regions, resulting in 288 sets of weighted ensemble means discussed in the paper. The differences between these setups are visualized, described and discussed. In the second part of the manuscript several recommendations are given regarding how to apply weighting methods in general.

With this manuscript the authors set out to answer a big question as stated in their title: 'to weight or not to weight'? And in more detail (line 77): 'Should model weights be developed separately when investigating different climate variables? Should model weights be estimated separately when investigating different domains?

I would like to propose a somewhat provocative argument about this aim: With the setup suggested here it is impossible to answer these questions. To advocate for (or against) weighting future projections in whatever way one would need to show an added value of the weighted ensemble compared to the unweighted one (for example increased skill by some metric). This is notoriously hard to prove (some would say impossible) as we do not know the ground truth in the future. Approaches have been suggested to circumvent this problem, at least partly. These include out-of-sample validation of weighted ensembles in the historical period where there are still observations available or model-as-truth approaches. None of this is done in this manuscript. The authors merely provide an extensive comparison of the effect of different weighting setups. As far as I can tell, most of the recommendations on weighting provided in the second part of the manuscript are not connected to the results presented (which mainly show relative difference between the different methods employed and as such can not answer the questions posed).

*Thanks again for these comments. The aim of this study is not to show that we find the 'best' future climate change projection, but to highlight the different approaches to estimate model weights and the resulting effects on the estimates of projected climate change and provide an extensive comparison of the effect of different weighting setups. The added value is shown in the reduction in bias in the historical period (e.g., Figure S3), and in the quantification and ultimately the reduction of uncertainty in the estimated climate change signal (e.g., Figures 5 and 6).*

*We thank the reviewer for pointing out that our recommendations are disconnected from the results. In this study, we show that, yes, for different variables and domains that different weights need to be estimated. And this ultimately produced different climate change signal. Although our results do not directly provide a way forward for model weighting (which is an extremely difficult problem to solve), this extensive study showcases how different strategies can impact estimated model weights and their respective climate change signals, which makes this a study that has not been done in this magnitude and provides comprehensive guidance for future studies and impact assessments which are considering incorporating model weighting. We have revised significant portions of the paper, particularly in Section 4, to better connect the results of the analysis to our recommendations and findings.*

My second main criticism is that the results presented in this work are not really new or surprising. The authors basically show that weights based on different variables and regions differ - but this is what they are designed to do. If weights for different regions and variables were all identical there would be something wrong with the model ensemble or the setup of the weights, right? Finally, the authors give several recommendations but these are more of a general nature and I had a hard time connecting them to the specific results presented. As a matter of fact several of the arguments have been made before and are not connected to any of the work done here (for example the discussion about spatial coherence in line 363f).

*We again agree with the reviewer that our recommendations can be better connected to our results. The primary purpose of this study was to provide the extensive and comprehensive comparison of the setups associated with multiple weighting strategies in a manner and extent that, to our knowledge, has not been done before. This is discussed more in our response to the*

*following comment. We have revised the manuscript to connect the specific recommendations more carefully to the results of this study.*

*Given the current debate on model weighting in the community, and the general sense of not knowing a path forward, an extensive and comprehensive study like ours is just what the community needs right now. Even though there is no direct path forward that is reported in our study, we provide an extremely large experimental matrix that other authors and scientists can draw on when asking for their own application whether "to weigh or not to weigh". In addition, the conclusions include general recommendations based on the author's experience and some additional specific recommendations more clearly tied to this study have been included.*

In addition, I find the heavy self-citation, partly ignoring large chunks of other literature, employed in this paper somewhat strange. I would encourage the authors to put their work better into the context of the international scientific literature (for example lines 35, 56-67, specific comments on lines 321, 325, 380). In addition, the authors state at several points that their study is the first to 'assess the sensitivities of the model weights and resulting ensemble means to the combinations of variables, domains, ensemble types (raw or downscaled), and weighting schemes used' (e.g. line 284). This might be so but what is the gain? Again, I am not surprised that the selection of the metrics used to inform the weights has an influence on the weights. If that was not so, weighting would hardly make sense, right?

*Our own work is included in this paper because those other studies are highly relevant to this current effort. However, we also refer to over a dozen other studies in the broader literature that specifically address model weighting as well:*
*Sanderson et al. 2015, 2017; Knutti, 2010; Knutti et al. 2017; Weigel et al. 2008; Pena and Van den Dool, 2008; Min and Hense, 2006; Robertson et al. 2006; Shin et al. 2020; Brunner et al., 2020ab; Kolosu et al. 2021; Skahill et al., 2021. We have incorporated additional studies to bolster our study further.*

The number of sets of weights (48) and the number of weighted means produced (288) is in my opinion too excessive. The authors should pick a few representative and/or interesting examples to discuss and move the rest of the results into the supplement. I found it almost impossible to follow the discussion of methods, domains, variables and ensembles that are in turn applied to ensembles and domains.

*We thank the reviewer for this recommendation. However, as mentioned in the response to a previous comment, the purpose of this paper is to provide an extensive and comprehensive comparison of the effect of different weighting setups and allow others to assess the question "to weigh or not to weigh" in their own application. As such, the extensive collection of weighting schemes and strategies is critical to include. In addition, the main text includes only a subset of the results, and other results are included in the supplemental materials.*

Finally, I would like to urge the authors to provide at least a basic description of the methods which are at the core of this manuscript. As it is, the reader is merely referred to three papers by the authors (Wootten et al. 2020a, Massoud et al. 2020a, 2019) for more information. For a potential reader (or reviewer) it would be quite convenient to have a more self-sustained paper with at least the basic setup of the methods clearly described and only the details requiring reading several more papers.

*We agree and the basic equations and explanation of the setup of the different model weighting strategies in the supplemental material for the readers and reviewers. There is also more detail on the weighting schemes added to section 2.4.*

Overall this manuscript has several major problems raised above beside the many specific issues outlined below and I do not think that it can be published without a major overhaul. This should include, most importantly, clearer formulated research questions that can be addressed in the manuscript and a clear separation between conclusions based on results and general recommendations based on the authors experience. In addition, a better representation of already existing literature and more focused plots (showing only a subset of cases) would help the manuscript.

*We thank the reviewer for the comment. We have made a clear distinction in the revision for specific recommendations based on this paper and general recommendations from the authors experiences. We have also included different referenced papers throughout the text, where relevant. Furthermore, the main manuscript represents only a subset of the figures that are deemed necessary to tell the story of this paper, and all the additional figures and analysis are provided in the supplemental material.*

Minor comments

title: the quite narrow focus on parts of the United states should be reflected in the title.

*The title now reads "To weight or not to weight: assessing sensitivities of climate model weighting to multiple methods, variables, and domains in the south-central United States"*

line 16: At this point I am confused about the terminology. My a priori assumption is that there are different weighting schemes and in addition each scheme might use different variables to calculate the weights. Here they are mixed up so either the authors use another terminology (then they should make it clear) or this should be reformulated.

*This sentence now reads "Results suggest that the model weights and the corresponding weighted model means are highly sensitive to the weighting scheme that is applied."*

16-21: I am not sure what the authors point is here as this behaviour seems to be totally expected? Is the important point not rather that the metrics (including variable and region) the weights are based on need to be well-justified? With cherry-picked metrics it is probably possible to achieve any kind of weighting, right?

*When applying model weighting to future climate projections, it is unclear how the estimated model weights will impact the resulting projected climate change signal. This is the reason for the broad application in our study, and two clearly different examples are listed in the abstract to highlight this difference.*

28: please introduce NCA

*Thank you, this now included.*

line35: can the authors please cite a broader sample of the literature not limiting it to their own publications (assuming that they are not the only ones publishing on that topic)?

*We that the reviewer for noticing this, and we agree. We have included other references here and throughout the manuscript with other works that are relevant to this topic.*

39: I would argue that the ensemble mean is not representative for the members (one of the reasons why we need weighting)

*The sentence in question is revised to the following: "Large and local scale assessments can make use of the entire ensemble of climate projections (composed of global climate models [GCMs]), or make use of the unweighted ensemble mean."*

44: model weights themselves can not have any skill I would argue

*Thank you, this statement now reads: "Projections based on model weights derived from historical skill have been shown to have greater accuracy than an arithmetic multi-model mean in many cases, provided that there is enough information to determine a weight for each model."*

47: As a matter of fact the idea of independence weighting has not only come up in the last few years and is, e.g., mentioned in Knutti 2010 which is cited by the authors in the line before.

*We thank the reviewer for this comment, this now reads 'more recently'*

60: 'performance skill of atmospheric rivers globally' again, what would be the skill of an atmospheric river? I assume the authors refer to the model skill in simulating atmospheric rivers?

*Thank you, this statement now reads 'performance skill of the models to simulate atmospheric rivers globally.'*

69: Knutti et al. 2017 did not base their weights (only) on precipitation as seems to be suggested here

*Thank you, this sentence now reads: "Some studies have applied model weighting to a certain variable or to multiple variables and went on to investigate climate change impacts for other variables (e.g. temperature or streamflow) (c.f. Knutti et al., 2017; Massoud et al., 2018)."*

70: What is a common variable?

*This sentence now reads: "The National Climate Assessment had previously considered weighting based only on commonly used climate variables (e.g. precipitation and temperature, Wuebbles et al., 2017), but discussions to use additional variables are currently ongoing."*

73: 'Other studies have applied model weighting to a specific domain (e.g. globally) and went on to apply the developed weights on a different domain (e.g. North America or Europe) (Massoud et al., 2019).' This sentence does not seem to make sense. Do the authors mean that they have calculated the weights based on metrics in one domain and then applied them to projections for another domain? Please reformulate this to make in more clear.

*The sentence now reads: "Other studies have calculated weights based on metrics in one domain (e.g. globally) and then applied them to projections for another domain (e.g. North America or Europe) (Massoud et al., 2019)."*

79: I am not convinced by the relevance of these research questions and their implications. For a weighting method to have skill the weights need to be based on metrics that are physically and statistically connected to the variable that the weights are applied to. See for example the discussion about emergent constraints in Hall et al. (2019; 10.1038/s41558-019-0436-6). In lack of a certain variable in a certain region that is informative for all other variables in all other regions the answer to both questions has to be yes, without any further analysis from a purely skill-based perspective I'd argue. There might be other considerations against it but they depend on the application (and are, hence, independent on the outcome), such as physical and spatial consistency of the weighted distributions.

*Thank you for this comment. We believe the reviewer is referring to the physical connection between temperature and precipitation here. This study deliberately constructed the experimental matrix to examine the sensitivity of different model weighting strategies (i.e.*

*including weighting on temperature and estimating future precipitation or vice versa) precisely to build toward addressing some of the concerns the reviewer mentions above. In addition, the authors of this study are all directly and indirectly involved with the discussions for the Fifth National Climate Assessment (NCA), led by the United States Global Change Research Program (USGCRP), surrounding downscaling and multi-model ensemble. There are 10-15 people representing multiple agencies of the United States government discussing issues associated with downscaling and model weighting during bi-weekly meetings over the past two years. The effort in this study, and the questions of interest in this study, delivers research on questions of interest to the USGCRP and the broader discussion group. This includes questions on the sensitivity of model weighting strategies (such as weighting on temperature and estimating precipitation, or vice versa).*

83: is the entire domain the combination of Louisiana and New Mexico or are there additional regions not covered by them? Maybe indicate the sub-domains in figure 1?

*Labels for the states used have been added to Figure 1. In addition, the sentence in question is revised to read as the following: "Furthermore, we use two sub-domains, the states of Louisiana and New Mexico, alongside the south-central U.S. study region."*

84: can the authors motivate why they use CMIP5 instead of the newer CMIP6?

*The following has been added to Section 2.2: "CMIP5 GCMs are used in this study because LOCA downscaling with CMIP6 was not available at the time of this writing."*

line 106/figure 1: I am not familiar with the term 'high temperature' is this the same as 'maximum temperature' which is (in my opinion) a frequently used term? And what is annual high temperature? Is it the maximum over different annual mean temperatures or the maximum of the maximum daily temperature or something else entirely? Over which time period?

*The sentence in question now reads: "Average annual precipitation in the southeast portion of the domain can be eight times higher than drier western locations and average daily high temperatures can reach 40°C (Figure 1)." The caption for Figure 1 has also been modified to match.*

115: Just so that I understand correctly, also the CMIP5 models are interpolated to 10km – corresponding to a resolution much finer than the native one?

*Yes, this sentence now reads as: "To facilitate analysis, the data for each ensemble member and the gridded observations are interpolated from their native resolution to a common 10 km grid using a bi-linear interpolation similar to that described in Wootten et al. (2020b)."*

section 2.4: The authors aim to provide a comparison of different weighting schemes but here these weighting schemes are not introduced at all requiring the reader to read several other papers to get any information at all about them. Please provide at least the basic properties and differences between the schemes investigated in this study.

*We agree, and the basic equations and explanation of the setup of the different model weighting strategies are provided in the supplemental material. More detail on the weighting schemes is also included in section 2.4.*

141: 'The Skill strategy utilizes each model's skill in representing the historical simulations' I assume the authors mean 'historical observations' here?

*The reviewer is correct, 'historical simulations' has been corrected to 'historical observations.'*

150: If the authors write 'weighting schemes are applied' here they mean that weights are calculated is that correct? I find this confusing since they also write 'applied' for the process of calculating a weighted mean of the future projections. Could the authors try to find a less ambiguous language throughout the manuscript?

*Yes, we added more clarification of the difference between weighting scheme and weighting strategy throughout. The first portion of section 2.5 now reads: "Each weighting scheme (Skill, SI-h, SI-c, and BMA) is applied to both ensembles (CMIP5 and LOCA) and three domains (south-central U.S., Louisiana, New Mexico) to fill out an experimental matrix of weights, representing a collection of weighting strategies. As a result, for each weighting scheme (skill, SI-h, SI-c, and BMA) and ensemble (CMIP5 and LOCA), there are six sets of weights produced (i.e. 3 regions and 2 variables). One example of a weighting strategy would be the BMA weighting scheme used on the CMIP5 ensemble trained on tmax for the entire domain. Another weighting strategy example would be a skill-based weighting scheme used on the LOCA ensemble trained on precipitation in Louisiana."*

156: '(ensemble choice x weighting methods choice x variable choice x domain choice = 2 x 2 x 3 x 4 = 48).' This is mixed up please correct

*Yes, ensemble choice x variable choice x domain choice x weighting methods choice = 2 x 2 x 3 x 4 = 48. This is corrected in the text.*

166: I am not sure I understand why the weights are applied to the sub-domains separately. The resulting maps should identical to the corresponding region in the full region, correct?

*The reviewer is correct that some will be identical. The study makes this point on lines 235-238: "However this maximum number of ensemble means resulting from the experiment contains several duplicates. For example, when using the same set of weights, the resulting ensemble mean in a subdomain will be the same as the resulting ensemble mean from the same portion of the full domain. As such, the actual number of ensemble means is smaller than 288." The following has been added at the end of the section to clarify: "However, we also note that there will be several duplicates in the experiment. For example, when using the same weighting strategy, the resulting ensemble mean in a subdomain will be the same as the resulting ensemble mean in the same portion of the full domain."*

figure 3: 'grey dots' do the authors mean the red dots?

*Yes. The figure caption has been adjusted accordingly.*

figure 3: as a general question: should weights not be normalized in order to be comparable across the different cases?

*Yes, the weights are normalized, which is why each y-axis goes up to 1. This has been clarified in section 2.4.*

182: 'One observation seen in these weighting combinations is that the weighting schemes themselves are all sensitive to the ensemble, variable, and domain for which they are derived.' I do not agree with this statement in this general form, could the authors provide a bit more detail? To give just two examples: the bcc model gets consistently low weights for all cases and the low weight of NorESM1 (among many other models) is not sensitive to variable and domains but only to the ensemble.

*This is true for models that get low weights consistently, but the comment in our study refers to which models that might have higher weights in some strategies but lower weights in others, and these are in effect the models that provide information to the future projections. This sentence now reads: "One observation is that the weighting schemes themselves are all sensitive to the ensemble, variable, and domain for which they are derived in terms of which GCMs are given the highest weight."*

185: what are 'model combinations'

*The highest weighted models that result from each weighting strategy is listed in Table 1. This sentence is revised to read: "From Table 1, no model appears in the top three for all weighting strategies."*

189: is this surprising given that (from what I understand) BMA is a structurally different method while the other three are variants of the same method?

*Yes, that is true. We will include a statement to point this out.*

193: I would tend to say the colour is red not orange. How is significance established for this case or is this just a qualitative statement? Then maybe use a different wording.

*Agreed, we should use a word other than significantly. This is changed to 'noticeably'.*

195: what are differences 'within each combination of ensemble, variable, and domain'?

*This sentence was deleted in response to the following comment.*

197: what does 'combinations' refer to here?

*Combinations refers to each weighting strategy (i.e., the weighting scheme and domain, variable, and ensemble used). We have revised this to refer to them explicitly to weighting strategies.*

206 'Similar to the CMIP5 ensemble in Figure 3, the BMA weights tend to be larger for the highest weighted models in the LOCA ensemble compared to those derived with the Skill, SI-h, and SI-c schemes' Can the authors speculate on the reason for this behaviour?

*The following is included in the text to speculate on this behavior: "We speculate that the reason for this is because the Skill, SI-h, and SI-c strategies involve the 'skill' of each model when estimating weights, and since the LOCA downscaled ensemble is bias corrected, most models have similar skill and therefore similar weights."*

212: 'the weights for the LOCA ensemble [tmax, Louisiana] generally range from 0.025 to 0.05' Do the authors mean 0.25-0.5? Otherwise it is impossible to see this in the figure 4. The authors might want to explain the notable exception from this. How is 'BMA best' calculated from the 100 iterations of BMA? How is a case like MIROC with a median of about 0.25 but a best of close to 0 possible?

*This means that, generally, most model weights for the LOCA/tmax/Louisiana strategy are between 0.025 and 0.05. For the second question, the MIROC model has a distribution of weights from BMA that includes lots of high values, but when sampling the combination that*

*produces the 'best' simulation (i.e. lowest bias compared to historical observation), the sampled combination of model weights just happens to be very low for MIROC. The following text is added at Lines 332-335: "The BMA best combination is the single set of model weights from the BMA posterior that creates a weighted model average that has the best fit to the observations. Although all the samples of model weights from the BMA posterior have an improved fit compared to the original ensemble mean and provide a range of model weights as shown in the BMA distributions in Figures 3 and 4, the BMA best combination is considered the best of all these samples."*

223: what is 'co-dependence between models in an ensemble'? Does 'Skill' account for dependence at all as seems to be suggested here?

*Co-dependence means when two models provide similar information to an ensemble average. The 'Skill' method does not consider co-dependence, and we will remove this strategy from this sentence. The 'SI-h' and 'SI-c' methods consider co-dependence in the historic and future simulations respectively. The BMA method down-weights models that provide similar information. See supplementary section of Massoud et al., 2020a that discusses this concept in more detail. More detail on this has also been added to section 2.4 of the manuscript.*

225: 'BMA tends to be the most sensitive' could this somehow be quantified?

*This is observed visually in Figures 3 and 4 in particular, where the magnitude and variability of the weights is much larger for BMA between the different variables, domains, and ensembles, then for the other three weighting schemes. We will clarify this in the text.*

239: So why not just not use the sub-domains at all?

*The results are only identical for a small sample of these different tests, but they are still unique for most examples.*

figure 5: is there are particular reason for selecting a base period of 25 years and a future period of 30 years? What do the boxes, whiskers represent?

*There are several other projects in the study region that use 1981-2005 as the historical period (including Wootten et al. 2020b). As such, this historical period was used to facilitate comparisons. This is explicitly stated in Section 2.2. Second question, the boxplots in Figure 5 represent the inter-model spread for both variables for both ensembles. The left-hand boxplots represent this for the historical period, and the right-hand side represents the projected changes from both ensembles.*

271-281: I am not sure I understand why this paragraph is here? Should the reader look at and understand all the figures listed here? Or is this just an outlook? The authors might want to consider dropping it.

*This section is an outline of the following sections to help guide the reader. It also points out which extra analyses and figures are included in the supplemental material. This section is left in to guide the reader.*

321: Maybe the authors could give some examples of the literature that does exist? To give just a few examples (there are more): 10.1029/2019GL083053, 10.1088/1748-9326/ab492f, 10.3389/frwa.2021.713537, 10.1029/2020JD033033

*In response to the major comments of both reviewers, Section 4, the Discussion section, was significantly restructured. The sentence where this comment was made was deleted. The references the reviewer provided are referenced in the introduction in Section 1.*

325: Again, there are counter-examples that might be good to mention here: 10.5194/acp-20-9961-2020, 10.3389/frwa.2021.713537

*In response to the major comments of both reviewers, Section 4, the Discussion section was significantly restructured. The sentence mentioned in this comment was removed as a result. The references the reviewer provided are incorporated in Section 4.4 of the revised manuscript.*

327: 'Third, for situations where projections are provided to impact models, does this type of study need to be repeated using impact model results' I don't think I understand this question.

*Thank you for this comment. In response to the major comments of the reviewers, the manuscript was restructured, and these research questions were stated more clearly in the introduction. This question is revised to the following: "Should a sensitivity analysis with multi-model weighting strategies be repeated using impact model results?"*

334: This is not correct so generally, see references above.

*As mentioned above, Section 4 was significantly restructured. The sentence mentioned in this comment was revised to the following and appears at the start of Section 4.3: "At the time of writing, discussion surrounding the use of weighted multi-model ensembles has been traditionally limited to climate model developers and the production of national or international climate assessments, but is beginning to be used in impact assessments."*

342: Who are these 'others'? Please provide references

*This statement now reads "Based on expert discussions surrounding downscaling and model weighting, the NCA is now considering weighting based on model climate sensitivity as opposed to traditional model weighting approaches."*

349: Why does a unweighed mean over-favor certain models? I would assume that by definition in an unweighted case all models are treated equally.

*You are right, what is meant here that certain models are provided higher weights than they should be receiving. This sentence now reads the following: "An unweighted mean will allow models with large biases and co-dependencies regardless of the domain or variable of interest larger influence in either climate models or impact assessments."*

354: applying multiple methods as suggested here might lead to contradictory results, can the authors say something about what a user that tries to get a single answer should do in such a case?

*Thank you for this question. That is the point we are reaching in our study, there is no 'single answer' and if the user wants a true accounting of the uncertainty to the question at hand, then the user should use many strategies if it is feasible to do so. This point is emphasize in the revised manuscript.*

380: 'Climate model evaluations and national assessments typically focus on the continental United States or North America.' There are assessments also for other continents.

*Thank you, this is corrected at the beginning of section 4.*

394: Is this recommendation somehow connected to the results shown in this manuscript or just the authors opinion?

*This reflects a combination of both the results in the manuscript and the authors experience. The manuscript has been revised to delineate connections more carefully between our recommendations and results in the manuscript.*

346: 'a multi-model ensemble of climate projections should incorporate model weighting' The ensemble itself can not incorporate weighting I'd argue. Weights can only be applied once the ensemble is aggregated along the model dimension (for example by calculating a multi-model mean).

*The statement the reviewer is referring to is on line 436-437. The reviewer is correct, the sentence now reads "...efforts using a multi-model ensemble of climate projections should incorporate model weighting."*

446 (recommendations): Could the authors connect these recommendations to their results?

*Thank you. Section 4 has been restructured to connect the recommendations to the results of analysis. We have also connected the recommendations to our numbered questions in the introduction.*

456: how can a domain be small compared to internal variability?

*Thank you for this question. What we mean is that the spatially aggregated internal climate variability of a smaller region is much larger than that of a larger domain, which make the model averaging results less coherent for a smaller domain as they would be for a larger domain. The extreme case of this is applying model weights using a single grid cell. This sentence is edited to the following: "First, this study makes use of domains that are fairly small where the spatially aggregated internal climate variability is larger than that of a large domain."*

**Citation**: https://doi.org/10.5194/esd-2022-15-RC2

---

## Author Response (AR2)

Responses to Reviewer 1

*Author Comments*: We thank the Reviewer for taking the time to review our manuscript a second time. Our responses to each comment follow the written text of the reviewer.

Review of "To weight or not to weight: assessing sensitivities of climate model weighting to multiple methods, variables, and domains in the south-central United States" by Wootten et al.

This is my second review of this manuscript. First of all I would like to thank the authors for their extensive answers to my comments and their work on the manuscript. I think I now understand better what they want to achieve with this study. However, I still see a mismatch between the answers the authors give to their research questions and the actual analysis done. In the first round of revisions I commented that the author's analysis is not sufficient to properly answer the posed research questions and this criticism still holds.

As I see it the manuscript has two parts:
1. an (extensive) analysis of weighting scheme impacts presented in section 3. and
2. recommendations in section 4 .

Both parts individually are mostly fine (for example I generally agree with most of the recommendations given – just not with the way they are intended to be based on the analysis). I don't really know how to resolve this larger issue I see with the manuscript other than focusing either only at 1. (and doing a purely physical analysis of weighting effects) OR only at 2. (writing a perspective style manuscript with general recommendations).

If the authors want to do both they need to cleanly connect both, clearly showing a chain of analysis-result-interpretation-recommendation. This is not done at the moment and therefore I can not recommend this manuscript for publication at this point.

*Author Comments*: We thank the reviewer for the comments and thorough review of our manuscript. In line with the comments above we have made significant changes to the manuscript. The new version of the paper now focuses on the extensive quantitative analysis of weighting strategies (which includes the weighting scheme and the choices surrounding variable[s] and domain of interest). The second portion of the manuscript, which used to include a qualitative perspective with general recommendations, is now removed from the paper. For this reason, we think that several of the comments below no longer apply with this revision.

Major comment on the analysis / Discussion from the last round of revisions

In answer to my comment on a missing skill analysis the authors write in their answer 'The added value is shown in the reduction in bias in the historical period (e.g., Figure S3), and in the quantification and ultimately the reduction of uncertainty in the estimated climate change signal (e.g., Figures 5 and 6).'

I am not convinced by either of these arguments:
- 'reduction in bias in the historical period': ultimately (I assume) the weighting is mainly

intended to improve assessments of the future and a reduced bias in the historical period (in sample) does not necessarily mean better model performance in the future (e.g. Sanderson et al. 2017). This is why I brought up the perfect model test.

- 'reduction of uncertainty in the estimated climate change signal': A reduction of model uncertainty does not necessarily mean a better representation of future climate. Indeed, the opposite could be the case if the raw model distribution was already overconfident (in this case an increase in uncertainty would be beneficial at least from a reliability perspective).

In figures 3 and 4 we see that the weighting strategies effectively reduce the model ensemble to only 3-5 models with non-zero weights. I wonder if such a reduction does not indeed rather lead to worse, i.e. overconfident/too narrow future projections?

The authors go on to write 'In this study, we show that, yes, for different variables and domains that different weights need to be estimated.'

I am sorry but this is not shown. The authors merely show that the weights differ for different cases. There are some figures showing the change in RMSE in the historical period but they are obviously not the focus of the analysis as they are placed in the supplement and not really discussed (figures S3-S6). And looking at them I can find some examples where using information from the same region leads to better historical RMSE scores but there are also counter examples to the overall interpretation remains unclear.

*Author Comments*: In response to comments from Reviewer 2, Tables 2-4 are added to the text to incorporate a clear discussion of the improvements offered by weighting strategies that also clarifies that there are differences in the improvement offered dependent on the domain, ensemble, and variable used. The following is added prior to the Discussion on Lines 413-425: "Figures S3-S6 indicate that all the weighting strategies used in this study resulted in higher skill for both high temperature and precipitation in all three domains. To summarize the results for skill, the RMSE of each weighting strategy is shown for all three domains for precipitation and high temperature in Table 2 and Table 3 and the RMSE for the unweighted cases are in Table 4. Of the weighting strategies using the CMIP5 ensemble 92%, 92%, and 75% have lower RMSE for precipitation than their unweighted counterparts for the full, New Mexico, and Louisiana domains. Similarly for the high temperature, 96%, 100%, and 79% of weighting strategies have lower RMSE than their unweighted counterparts for the full, New Mexico, and Louisiana domains. Therefore, most weighting strategies have higher skill than the unweighted CMIP5 ensemble. However, there is a similar pattern for weighting strategies using the LOCA ensemble. For precipitation, 79%, 58%, and 67% of weighting strategies using the LOCA ensemble have a lower RMSE than their unweighted counterparts for the full, New Mexico, and Louisiana domains. Similarly for high temperature, 88%, 88%, and 83% of weighting strategies using the LOCA ensemble have a lower RMSE than their unweighted counterparts."

Major comment on the recommendations

In the abstract the authors write 'From the results of our analysis, we summarize our recommendations concerning multi-model ensemble weighting as follows: […]' followed by a list of recommendations. This to me clearly indicates that the recommendations are based on the analysis and I (still) disagree with that to a large extend. A similar situation arises in section 4

but for brevity I will focus mainly on the abstract here to make my point.

- 'That model weighting, if used, be derived using both common (e.g., precipitation) and stakeholder-specific (e.g., streamflow) variables [...]'
How can this statement follow from the analysis if the authors do not look at any stakeholder-specific variables in their analysis?

*Author Comments*: Thank you. This comment has been removed from the text.

- 'That weighting is derived for individual sub-regions in addition to what is derived for the continental United States [...]'
I completely agree with this statement from an expert perspective. But again, how is this derived from the results presented in section 3?
I assume some of the results shown in the RMSE figures S3-S6 could be used to make this case but this is not done in the manuscript.

*Author Comments*: This comment has also been removed from the text.

- 'Multiple strategies for model weighting are employed when feasible, to assure that uncertainties from various sources (e.g., weighting strategy used, domain or variable of interest applied, etc.) are considered.'
I find it hard to support this statement so generally. My argument would rather be that each weighting strategy used (and I agree that using multiple can be advantageous) should be justified. Just using multiple strategies with weights based on various regions and variables does not lead to any obvious benefits I can see if not supplemented by a more in-depth analysis of method skill or system understanding.
To connect this to the analysis take for example figure 8: here weighted mean values from 24 different strategies are presented. But what is the added value of this alone? There are quite considerable differences in the weighted means depending on the strategy and they cover both, higher and lower values than in the unweighted case. This actually leaves me with the slightly uncomfortable impression that it might be better to just use the unweighted case.

*Author Comments*: As mentioned above, these recommendations were removed from the paper. The paper now focuses on purely a quantitative analysis of the extensive results. As such, we hope this removal of text satisfies the concerns mentioned above by the reviewer.

Minor comments

17: I am sorry but I still find the convention used here unclear. The authors write that '...model weights and the corresponding weighted model means are highly sensitive to the weighting scheme that is applied'. But then the examples given here do not address different weighting schemes but only differences in the weights when they are based on different regions.

*Author Comments*: Thank you for this comment. We corrected this sentence in our abstract to read 'weighting strategies' instead of 'weighting scheme', and we go on to explain on Lines 127-132:

"For reference for the reader, we define weighting schemes to refer to the numerical approach to weighting alone, such as Bayesian Model Averaging (BMA) or the approach defined by Sanderson et al. (2015, 2017). We define a weighting strategy as the weighting scheme and other choices made when using the weighting scheme to derive model weights. For example, a weighting strategy would be using the BMA weighting scheme to derive weights using the continental United States and daily high temperature alone and another weighting strategy would be using the BMA weighting scheme to derive weights using the Southern Great Plains of the United States and daily precipitation alone. Both such examples use the BMA weighting scheme, but with different choices made to derive weights, making the two examples different weighting strategies."

- 'when estimating model weights based on Louisiana precipitation, the weighted projections show a wetter and cooler south-central domain in the future compared to other weighting schemes" Do the authors mean 'other regions' instead of ' other weighting schemes'? Otherwise I do not understand the sentence.

*Author Comments*: In line with the previous comment, we corrected this sentence in our abstract to read 'weighting strategies'.

- 'Alternatively, for example, when estimating model weights based on New Mexico temperature, the weighted projections show a drier and warmer south-central domain in the future. However, when considering the entire south-central domain in estimating the model weights, the weighted
future projections show a compromise in the precipitation and temperature estimates.'

My viewpoint is that there are, one the one hand, different weighting schemes and, on the other hand, these weighting schemes can be used to calculate weights based on different variables and regions. But in the second case (at least for me) it is still the same weighting scheme. This is what I tried to convey with my comment on the same topic in the last round of revisions and in fact the authors themselves make the same differentiation later (line 89) but not here it seems.

*Author Comments*: Thank you again. The point the reviewer made is correct, in that the different weighting schemes can be applied to calculate weights based on different variables and regions, and this is what a weighting strategy is. We have added a statement to clarify this in the introduction on Lines 127-132 to make our definition clear for the reader.

327: 'The internal variability is represented by the ensemble average of the standard deviation of each variable from each ensemble member (per Hawkins and Sutton, 2009; 2011)'
Just to make sure I understand correctly what was done here: the authors have removed the estimated forced response using a 4th order polynomial and then calculated the time standard deviation as an estimate of internal variability as in the Hawkins and Sutton studies?

Maybe the authors want to mention the caveats of this approach? Mainly that the polynomial fit is not a very good estimation of the actual forced response (in particular regionally) as shown, for example, in a recent related publication using large ensembles: 10.5194/esd-11-491-2020

*Author Comments*: In this case, the calculation is over time, but we did not remove the estimated forced response since we could not estimate the polynomial regression from the historical period (1981-2005) all the way through the future period (2070-2099), as the period of the data was not continuous to make a robust regression. The sentence in question is revised on Lines 382-385 to read the following with an additional reference in support:

"The internal variability of the historical and future period is represented by the ensemble average of the standard deviation of each variable from each ensemble member over time (per Hawkins and Sutton, 2009; 2011, Maher et al. 2020) for each of the three domains (full, Louisiana, and New Mexico). However, we note that the forcing response is not removed given the temporal period is not continuous which is a caveat for this analysis."

Figure 8-13: This is just a suggestion but would plotting the change to the unweighted case in the small panels not show the effect of the weighting strategies better?

*Author Comments*: Yes, doing so may show more of the differences between the unweighted case and the individual weighting strategies in those Figures. However, we think that these figures together with Figures 6 and 7 (which show each weighting strategy compared to the unweighted case in a different format) make these differences apparent and allow comparisons between weighting strategies. As such, no change is made to Figures 8-13.

Responses to Reviewer 2

*Author Comments*: We thank the Reviewer for taking the time to review our manuscript a second time. Our responses to each comment follow the written text of the reviewer. There's a note that must be made at this point. In response to Reviewer 1 the manuscript was changed significantly to focus on the analytical analysis rather than the original questions and recommendations. For this reason, many comments made with respect to previous editions of this manuscript may no longer apply. We have responded to comments that remain in the manuscript and noted those that no longer apply below.

I continue to remain very doubtful about the value of this study. The authors have chosen not to address any of my main suggestions concretely, and they continue to defend the normative quality of this study, with a list of specific questions that they set out to answer on the basis of their results. Except, nothing in their results supports any of their answers to the list of questions, I'm sorry to say.

Once again, their results only confirm something we have known for a while, that different weighting schemes deliver different outcomes. I'm going to argue that nothing about this – expected – outcome supports the authors' recommendations. Specifically (the authors' words in italics)

*Author Comments*: We thank the reviewer for the comments. The new version of the paper now focuses on the extensive quantitative analysis of weighting strategies (which includes the weighting scheme and the choices surrounding variable[s] and domain of interest). The second portion of the manuscript, which used to include a qualitative perspective with general recommendations, is now removed from the paper.

That model weighting, if used, be derived using both common (e.g., precipitation) and stakeholder-specific (e.g., streamflow) variables to produce relevant analysis for impact assessments or using multiple climate variables relevant for a national assessment region (Question 1).
The authors don't use stakeholder-specific variables, or multiple variables, so nothing from their study shows the added value of doing that and therefore supports this first recommendation.

*Author Comments*: Thank you. This comment has been removed from the text.

That weighting is derived for individual sub-regions in addition to what is derived for the continental United States or other nations and that weighting for impact assessment is also derived for a domain relevant to the impact assessment (Question 2).
The authors results show that the outcome will be different depending on what domain is used. What would the final outcome of doing this multiple types of weighting be (besides once again discovering that the results vary)? Which one is right and which one is wrong?

*Author Comments*: This comment has also been removed from the text.

Weighted ensemble means should be used not only for national and international assessments but also for regional impacts assessments and planning (Question 3).
What, from the authors' results, support this recommendation? What would impact assessment and planning do with the results of weighting, which either will be fraught by the knowledge that the specific result is very sensitive to the actual choices made for the weighting schemes, or will be delivering multiple results in front of which the planner will be left with having to make a decision? What is the added value of these weighted schemes if we have no idea of their skills?

*Author Comments*: This comment has also been removed from the text.

Multiple strategies for model weighting are employed when feasible, to assure that uncertainties from various sources (e.g., weighting strategy used, domain or variable of interest applied, etc.) are considered (Question 4).
What concretely would the user do with the results of multiple weighting schemes? Again, why do this if nothing has demonstrated the added value of using a weighting scheme? I could argue that throwing out models at random will produce different results from these weighting schemes. Should that be done just to demonstrate that if CMIP had included different models we would have gotten different results? I would argue that is as important a recognition as any. But nobody sets out to do that for the sake of it.

*Author Comments*: This comment has also been removed from the text.

Future efforts should examine the weighting of impacts model outputs from climate model inputs (Question 5).
We don't know what to do with climate model inputs weighting…how are we going to combine that with impacts model weighting? Wouldn't it be better to first figure out what skills model weighting has, if any at all? And again, and for the last time, where is this recommendation coming from, on the basis of the paper results?

*Author Comments*: This comment has also been removed from the text.

Latching on to this last comment, I would like to underline that these recommendations would make any sense only if the authors had shown any skill in the weighted projections compared to the unweighted, but none of that is shown in this paper. I would find the publication of these recommendations actually problematic and harmful, not being substantiated in any way.

*Author Comments*: Thank you. Given the response to Reviewer 1 and the resulting changes to the manuscript many of the above comments no longer apply with respect to the manuscript. However, the reviewers' final comment above does merit a response. The reviewer claims that we did not demonstrate the skill of the weighted projections. Figures S3-S6 show the RMSE of the weighted ensembles compared to the unweighted case for each of the weighting strategies considered in the study. These four figures clearly show that the weighted projections have greater skill than the unweighted case for both high temperature and precipitation in all three domains. In this revised manuscript, we have noted this in the main manuscript and refer the reader to the Supplemental Figures.

Furthermore, Tables 2-4 are added to the text to incorporate a clear discussion of the improvements offered by weighting strategies that also clarifies that there are differences in the improvement offered dependent on the domain, ensemble, and variable used. The following is added prior to the Discussion on Lines 413-425:

"Figures S3-S6 indicate that all the weighting strategies used in this study resulted in higher skill for both high temperature and precipitation in all three domains. To summarize the results for skill, the RMSE of each weighting strategy is shown for all three domains for precipitation and high temperature in Table 2 and Table 3 and the RMSE for the unweighted cases are in Table 4. Of the weighting strategies using the CMIP5 ensemble 92%, 92%, and 75% have lower RMSE for precipitation than their unweighted counterparts for the full, New Mexico, and Louisiana domains. Similarly for the high temperature, 96%, 100%, and 79% of weighting strategies have lower RMSE than their unweighted counterparts for the full, New Mexico, and Louisiana domains. Therefore, most weighting strategies have higher skill than the unweighted CMIP5 ensemble. However, there is a similar pattern for weighting strategies using the LOCA ensemble. For precipitation, 79%, 58%, and 67% of weighting strategies using the LOCA ensemble have a lower RMSE than their unweighted counterparts for the full, New Mexico, and Louisiana domains. Similarly for high temperature, 88%, 88%, and 83% of weighting strategies using the LOCA ensemble have a lower RMSE than their unweighted counterparts."

---

## Author Response (AR3)

Response to Reviewer Comments

I acknowledge the efforts the authors seem to have put into revising their paper. However, I also have to stress that they seem to have ignored large parts of my first major comment from the last round of revisions (which in turn was already a follow-up from the first round).

We apologize that the reviewer believes that we ignored portions of their comments from the revisions. We would like to offer clarifications to address these concerns and suggestions for additions to further address the reviewer's remarks. Our comments in response to the reviewer are in blue and follow each section of reviewer comments.

It reads: "In answer to my comment on a missing skill analysis the authors write in their answer 'The added value is shown in the reduction in bias in the historical period (e.g., Figure S3), and in the quantification and ultimately the reduction of uncertainty in the estimated climate change signal (e.g., Figures 5 and 6).'

I am not convinced by either of these arguments:
- 'reduction in bias in the historical period': ultimately (I assume) the weighting is mainly intended to improve assessments of the future and a reduced bias in the historical period (in sample) does not necessarily mean better model performance in the future (e.g. Sanderson et al. 2017). This is why I brought up the perfect model test.
- 'reduction of uncertainty in the estimated climate change signal': A reduction of model uncertainty does not necessarily mean a better representation of future climate. Indeed, the opposite could be the case if the raw model distribution was already overconfident (in this case an increase in uncertainty would be beneficial at least from a reliability perspective).

In figures 3 and 4 we see that the weighting strategies effectively reduce the model ensemble to only 3-5 models with non-zero weights. I wonder if such a reduction does not indeed rather lead to worse, i.e. overconfident/too narrow future projections?"

Again, we apologize that the reviewer believes that we ignored these remarks. This was not our intention. We very much agree with the reviewer that the reduced bias in the historical period does not mean better model performance in the future period. We also agree that if the raw model distribution is itself overconfident that increasing the model uncertainty would be beneficial.

On the first point, our intention with this manuscript was to assess the sensitivity of future projections to the various weighting strategies. We do not think that our historical weights justify using any or all the weighting strategies for the future projections. To the best of our knowledge, we have not said or implied this in our manuscript. However, we understand that our manuscript may not have gone far enough in discussing this point. We have included the following specific statements to this point at the end of Section 3 on Line 362 of the revised manuscript.

"It is important to note that this analysis of RMSE and bias is for the historical period only. Prior studies have noted that reducing historical biases does not mean better performance during the future period (Dixon et al. 2016; Sanderson et al. 2017). Therefore, historical skill alone does not justify the use of any weighting strategy. In what follows, we do not recommend using any specific weighting strategy based on the historical skill. Rather, we focus on the sensitivity of the projected changes to the various weighting strategies."

On the second point, we again agree with the reviewer that (theoretically) increasing the uncertainty would be appropriate if the raw ensemble is already overconfident. However, this is not the case in our analysis. The right-hand panels of Figure 5 in the manuscript show the spread of projected changes around the mean from the raw ensemble for both variables, while Figure 6 show the reduced spread of projected changes from the application of each weighting strategy, particularly from the BMA results. We recognize that we did not make this statement explicitly in previous iterations, but the reason for including these figures and the associated text was precisely to address this concern. The following is included at the end of Section 3.3 on Line 291:

"In addition, the right-hand panels of Figure 5 show that the projected changes around the mean from the raw ensemble are significantly larger than the reduced spread in Figure 6 (particularly from the BMA results) in the

weighted ensembles. This suggests that the raw ensemble has less confidence for both variables, both ensembles, and all three regions compared to the weighted ensembles."

On the final point above, the reviewer is correct that the several models regularly have large weights, but no models are given zero weight regardless of the weighting strategy used. We have provided the weights themselves in Table S2 in the supplemental material to clarify this. In addition, we have included the following text in the manuscript at the end of Section 3.3 on Line 296 to direct readers to the model weights.

"The weights for each model from each multi-model weighting strategy are given in Tables S2-S7."

We agree with the reviewer that having a few models with non-zero weights could make future projections overconfident, but this is unlikely in this case given that the spread of the raw ensemble for both variables was quite wide, even in the case of projected changes from the downscaled ensemble, as we have discussed above.

So, in short, my criticism was that the authors use the historical (in sample) RMSE improvement to justify weighting future changes by putting most of the weight on only a few models. I also pointed out that this might lead to overconfident results for projections of future climate.

Obviously it is, generally speaking, up to the authors to implement reviewer comments or argue against them. I think there might be arguments why implementing my comment is not relevant or feasible for this case but what the authors seem to do in their answer is simply reiterating their results.

Basically, they refer to additional tables and figures showing improvements in historical RMSE, if I am not mistaken. I am sorry, but this answer just does not address my comment. It is not surprising that historical bias is reduced when weighting based on historical bias but this does not justify (without further analysis and discussion) to apply the same weights to projections of future change.

Again, it was not our intention to ignore the reviewer's remarks and we apologize if we have inadvertently done so. To reiterate, the purpose of our manuscript is to examine the sensitivities of projected changes associated with weighting strategies. In a previous response to the reviewers (dated August 2022), we believed we had addressed the comment of the reviewer related to the aforementioned perfect model test and provided justification as to why we elected not to perform a perfect model test for this manuscript. The following is from the response to reviewers submitted in August 2022:

"While we agree that with the reviewer that this is a useful exercise, the question of the stationarity of weighting schemes is beyond the scope of this analysis and worthy of a manuscript in and of itself. In addition, the perfect model method assumes that the model that is chosen is a good approximation to the truth and that the future climate simulated in this model, along with the change in climate that occurs within its simulations, is representative of actual climate change and that the other models in the ensemble should have the same change signal. For example, in Brunner et al., (2019, ERL) it mentions 'For all regions there is also a chance that the skill decreases due to the weighting. This can happen if the perfect model has a very different response to future forcing compared to the other models, leading to the weighted multimodel ensemble moving further away from the 'truth'.' Furthermore, when applying weighting on the LOCA downscaled data, this perfect model test becomes irrelevant since all the models are bias-corrected to apply the downscaling. The author team is considering examining the question in a future study where we would also vary the model used as the absolute truth to examine some of the assumptions associated with the perfect model approach."

We recognize that the above may not have sufficiently answered the reviewers' concerns and suggestions and we have included the following language at the end of Section 3 on Line 362 to emphasize that we do not view the historical weights as justification for their future use.

"It is important to note that this analysis of RMSE and bias is for the historical period only. Prior studies have noted that reducing historical biases does not mean better performance during the future period (Dixon et al. 2016; Sanderson et al. 2017). Therefore, historical skill alone does not justify the use of any weighting strategy. In what

follows, we do not recommend using any specific weighting strategy based the historical skill. Rather, we focus on the sensitivity of the projected changes to the various weighting strategies."

In addition, the following language is included in the conclusions on Line 546 to note that further work should be done specifically on the question of the bias of future projections associated with multi-model weighting strategies.

"Future research will examine the accuracy and sensitivity using a perfect model exercise (such as what is described by Dixon et al. 2016 and Sanderson et al. 2017) to test the stationarity assumption associated with ensemble weighting. This is important since studies like Sanderson et al. (2017) show that a more skillful representation of the present-day state does not necessarily translate to a more skillful projection in the future. Our study does not consider the skill of the multi-model weighting strategies in the future projections, but rather it assesses the sensitivity of future projections to the various multi-model weighting strategies."

I find the lack of response to this point particularly striking because there is (among other work) a paper from Sanderson et al. (10.5194/gmd-10-2379-2017) which was done in the frame of the fourth National Climate Assessment for the United States (NCA4) which seems to be the same framework the authors work in. In the work from Sanderson et al. one can find statements like: "A more skillful representation of the present-day state does not necessarily translate to a more skillful projection in the future. In order to assess whether our metrics improve the skill of future projections at all, we consider a perfect model test where a single model is withheld from the ensemble and then treated as truth." So my criticism on the authors approach is not a personal opinion but reflected in the published literature and, in fact, in literature from the same climate assessment the authors work on.

Again, we apologize if the reviewer believes that we intended to ignore their remarks in previous reviews. We want to clarify that we don't think our historical weights are justified for the future. Instead, our paper shows the extensive matrix of results without choosing or recommending any specific weights. We understand that weights obtained using a historical bias do not necessarily mean a lower bias in the future. We are in complete agreement with the reviewer on this point. In our original edits, we tried to specifically address this comment with the following statement in the conclusions:

"Third, this study does assume stationarity in the multi-model ensemble weights and resulting weighted means. Future research will examine the accuracy and sensitivity using a perfect model exercise (such as what is described by Dixon et al. 2016) to test the stationarity assumption associated with ensemble weighting."

We acknowledge this might not be enough to address this concern, therefore we have extended the statement above to include the following additional statements inspired by the comments made by the reviewer:

"Third, this study does assume stationarity in the multi-model ensemble weights and resulting weighted means. Future research will examine the accuracy and sensitivity using a perfect model exercise (such as what is described by Dixon et al. 2016 and Sanderson et al., 2017) to test the stationarity assumption associated with ensemble weighting. This is important since studies like Sanderson et al. (2017) show that a more skillful representation of the present-day state does not necessarily translate to a more skillful projection in the future. Our study does not consider the skill of the multi-model weighting strategies in the future projections, but rather it assesses the sensitivity of future projections to the various multi-model weighting strategies."

We hope that including the above text will address the reviewer's concerns and show that we agree with the reviewer on this topic, and make it more clear to the reader what the actual aims of this study are.

So, given that I have now basically made the same comment twice and it has remained unanswered, I am sorry to say that my recommendation is now to reject this manuscript.

We apologize if we have ignored the reviewers' comments in our responses. It was not our intention to do so. To reiterate, the purpose of our manuscript is to assess the sensitivities of future projections to the numerous available options for weighting strategies. As our title states, we are "Assessing sensitivities of climate model weighting to

multiple methods, variables, and domains in the south-central United States". We completely agree that weights obtained using historical bias do not necessarily translate into a lower bias in future projections. We hope that the above comments address the reviewer's concerns.

---

## Author Response (AR4)

Response to Reviewers

*We thank the editor for the thoughtful comments to improve our manuscript. Our responses to each comment are included below in blue.*

**Comments to the author**:
Tracked changes version
Lines 42-44: This sentence structure is confusing me.

*This sentence has been deleted and the prior sentence now reads as the following: " More recently, weighting based solely on skill has given way to weighting based upon both skill and independence in recognition of differences in skill between regions and variables and the lack of independence between GCMs resulting from common bases in model structure (Massoud et al. 2019, 2020a; Sanderson et al. 2015b, 2017; Knutti, 2010; Knutti et al. 2017)."*

Lines 50-60: This is coming across as though you're trying to justify your own work. I'd rather see these citations raised in the context of scientific discovery and how the studies have advanced the field.

*This paragraph (Lines 49-56) now reads as: "Several studies have examined the effect of model weighting on the outcome of climate change projections from multiple ensembles. For example, in Massoud et al. (2019), the authors utilized information from various model averaging approaches to evaluate an ensemble from the Coupled Model Intercomparison Project Phase 5 (CMIP5; Taylor et al., 2012), finding that Bayesian Model Averaging (BMA) reduced the error by one third and constrained the uncertainty to 20-25% of the raw ensemble for projections of atmospheric river frequency. Massoud et al. (2020a) found that BMA constrained the uncertainty in precipitation projections over the contiguous United States (CONUS) to be a third of that in the original ensemble. In Wootten et al. (2020a), the authors found that ensemble weighting can change dramatically when weighting schemes are applied to statistically downscaled ensembles compared to a raw GCM ensemble."*

Lines 62-70: You're kind of just listing these studies and what they did. I'd like to see conclusions that motivate why you chose the problem that you're tackling in the present study.

*This paragraph (Lines 58-77) is now written as the following: "Other studies have applied model weighting to a certain variable or to multiple variables and went on to investigate climate change impacts for other variables (e.g., energy and hydrologic cycles). For example, Knutti et al. (2017) extended the weighting scheme of Sanderson et al (2015a; 2017) to projections of Arctic September temperatures and sea ice, finding that the uncertainty could be constrained by the scheme while noting the proposed weighting scheme is one of several that could be used for multiple applications. The National Climate Assessment had previously considered weighting based only on commonly used climate variables (e.g., precipitation and temperature, Wuebbles et al., 2017), but discussions to use additional variables, such as equilibrium climate sensitivity, are currently ongoing. Other studies have calculated weights based on metrics in one domain (e.g. globally) and then applied them to projections for another domain (e.g. North America or Europe) (Massoud et al., 2019). However, these studies are rare, as are studies providing comparisons of various weighting schemes. Examples of these studies include Shin et al. (2020), Brunner et al. (2020a), and Kolosu et al. (2021). Shin et al. (2020) suggested that researchers may provide results from several weighted ensembles to capture the uncertainties of future changes, but did not explore weighting strategies beyond different weighting schemes. Brunner et al (2020a) found that the region can influence the agreement between approaches to constrain uncertainty in the CMIP5 multimodel ensemble. Finally, Kolosu et al. (2021) focusing on a water-related decision context in Africa, finds that projected risk profiles were less sensitive to the weighting schemes used. Such studies as in these examples tend to focus on the sensitivity associated with one to a few components of a multi-model weighting strategy. No prior study (to the author's knowledge) offers a comprehensive cross-comparison of the sensitivity resulting from the choices of the domain, variable, weighting scheme, and ensemble that comprise multi-model weighting strategies. In addition, the primary focus of these studies are continental regions although climate projections are now being used by regional and local organizations for climate impacts assessments and climate adaptation with additional modeling efforts."*

Lines 81-83: I like that you're pointing out the novelty of the study. Please also explain why it's important to do this. That is, there are previous studies that have looked at individual pieces of this problem. Why is it important to look at them all together?

*The following sentences are added after this sentence (Lines 90-92) to address this: "Prior studies have examined some of these dimensions individually, but the comprehensive experimental matrix used here allows the comparison of model weighting results based on all these dimensions. This is important because there could be high sensitivities in the estimated model weights based on how the weighting strategy is formulated."*

Lines 86-87: I'd like to see a sentiment like this earlier in the paper. Saying that we don't have reliable local information for adaptation planning (and that it's kind of difficult to get, hence motivating your study) really points out the problem.

*The following has been added at Lines 75-77: "In addition, the primary focus of these studies are continental regions although climate projections are now being used by regional and local organizations for climate impacts assessments and climate adaptation with additional modeling efforts."*

Lines 102-105: I like this, but I'd also like to see something a little punchier. Perhaps a sentence after this one saying something like "That is, we want to figure out under what circumstances the projections are sensitive to weighting and why." I think you're getting at this in the final sentence of this paragraph (lines 107-109), but it could be clearer.

*To address this, the final sentence of the paragraph (Lines 116-118) now reads as: "Our purpose in this study is not to address the skill of the multi-model weighting strategies in future projections, but rather to assess under what circumstances the projections are sensitive to multi-model weighting strategies and why."*

Lines 122-123: It's good to include this, but also CMIP5 is a perfectly legitimate ensemble to use. Communities making adaptation plans aren't necessarily always going to use the latest and greatest, nor should they - imagine if they had used the CMIP6 archive before we figured out why some models had erroneously high climate sensitivities. A flexible method that can work on CMIP5 (or even CMIP3) is valuable.

*Thank you for pointing this out. To make this point we have added the following in section 2.2 at Lines 132-133– "That said, the weighting schemes used here are applicable also to other ensembles such as CMIP6 and CMIP3. Therefore, the findings of this study are generalizable to other ensembles."*

Lines 130-132: (I'm including this here because I'm reading in order. If you address this later, please ignore this comment.) I agree with getting a high signal-to-noise ratio. It might be worth a comment in the conclusions section about using other scenarios. For example, if you used RCP2.6, would your uncertainty ranges go down, meaning that the amount of climate change has a direct bearing on the sensitivity of the weighting scheme? (This also has implications for the time period that one is using - there may be less uncertainty for mid-century vs late-century projections.)

*Thank you for pointing this out. To address this point, we have added the following in Section 4.3 (Lines 509-512): "In this study, we focused on the sensitivity under RCP 8.5 to maximize the effects observed from different weighting strategies. Given the smaller change signals under other RCPs it is possible that the sensitivities observed here have a lesser magnitude under other RCPs. Considering this component is another aspect that could be explored in future work."*

Line 444: Did you mean that this has _not_ been explored? Otherwise I'd like to see some citations.

*Thank you for pointing this out. You are correct, we meant to say "has not been explored." This is corrected in the text.*